# Stealth edits to large language models

**Oliver J. Sutton**[*]
King's College London
oliver.sutton@kcl.ac.uk

**Qinghua Zhou**[*]
King's College London
qinghua.zhou@kcl.ac.uk

**Wei Wang**
University of Leicester
ww152@le.ac.uk

**Desmond J. Higham**
University of Edinburgh
d.j.higham@ed.ac.uk

**Alexander N. Gorban**
University of Leicester
a.n.gorban@le.ac.uk

**Alexander Bastounis**
King's College London
alexander.bastounis@kcl.ac.uk

**Ivan Y. Tyukin**
King's College London
ivan.tyukin@kcl.ac.uk

## Abstract

We reveal the theoretical foundations of techniques for editing large language models, and present new methods which can do so without requiring retraining. Our theoretical insights show that a single metric (a measure of the intrinsic dimension of the model's features) can be used to assess a model's editability and reveals its previously unrecognised susceptibility to malicious *stealth attacks*. This metric is fundamental to predicting the success of a variety of editing approaches, and reveals new bridges between disparate families of editing methods. We collectively refer to these as *stealth editing* methods, because they directly update a model's weights to specify its response to specific known hallucinating prompts without affecting other model behaviour. By carefully applying our theoretical insights, we are able to introduce a new *jet-pack* network block which is optimised for highly selective model editing, uses only standard network operations, and can be inserted into existing networks. We also reveal the vulnerability of language models to stealth attacks: a small change to a model's weights which fixes its response to a single attacker-chosen prompt. Stealth attacks are computationally simple, do not require access to or knowledge of the model's training data, and therefore represent a potent yet previously unrecognised threat to redistributed foundation models. Extensive experimental results illustrate and support our methods and their theoretical underpinnings. Demos and source code are available at https://github.com/qinghua-zhou/stealth-edits.

## 1  Introduction

The latest meteoric rise of artificial intelligence has been driven by the maturing of large language models. These models, predominantly based on the *transformer* architecture [39], have demonstrated remarkable abilities in natural language communication and comprehension, which are only beginning to transform the world as we know it today. Scaling has proved to be key in enabling these breakthroughs: the GPT-4 family of models, for instance, has on the order of $10^{12}$ trained parameters, a figure which would have been inconceivable only a few years ago. The raw computational power and vast quantities of quality data required to train models such as these make developing them prohibitively expensive for all but a handful of the world's wealthiest technology companies [31].

---

[*]These authors contributed equally to the paper and are listed alphabetically.

38th Conference on Neural Information Processing Systems (NeurIPS 2024).

This environment has also seen the rise of foundation models: collections of accessible or open source language models which have become an invaluable tool for those without the facilities to train their own from scratch. As language models have developed, 'hallucinations' — non-factual or non-sensical information generated by the model — have become challenging barriers to trustworthy and reliable artificial intelligence. Much work has been invested in trying to understand the origins of hallucinations [7] and develop mechanisms to mitigate them [18], amplified by regulatory requirements placed on organisations deploying AI by the European Union's recent 'AI Act' [9] or the UN's resolution on 'safe, secure and trustworthy artificial intelligence' [38]. Recent work has, however, shown that hallucinations may in fact be an inevitable artefact of fixed language models [19, 43].

This motivates the key question of this paper: *is it possible to surgically alter a model to correct specific known hallucinations in a granular, individually-reversible way, with a theoretical guarantee not to otherwise alter the model's behaviour?* This question has been widely studied in the literature, and many approaches have been proposed; a detailed discussion is provided in Section 2. Perhaps the approaches that come closest to answering this question are GRACE [15] and Transformer-Patcher [17]. GRACE selectively applies edits by comparing input features to a set of pre-computed keys. However, adding this to a model requires re-writing the model code, rather than updating existing weight matrices, and the conditional logic required does not naturally suit massively parallel computing architectures such as GPUs. Transformer-Patcher instead encodes edits into the last transformer perceptron block of a network, to detect when incoming features should be edited and provide the corrected output. Experimental studies have shown the potential of these approaches for targetted corrections to hallucinations, yet a clear theoretical understanding of what determines their success has remained elusive until now.

Here, we systematically study these and related methods under the collective banner of *stealth editing*. We develop a novel theoretical approach which reveals that, surprisingly, a single metric provably determines the editability of a given model (Theorem 2). The metric can be estimated directly from data and measures the *intrinsic dimensionality* of the model's feature vectors and was introduced in the setting of binary classification in [32]. Guided by this understanding, we are able to propose a simplified editing mechanism which optimises the selectivity of each edit. Through this, we are able to build a bridge between our methods, GRACE and Transformer-Patcher, showing that they can be implemented and studied within the same framework. The clear implication of this is that the new theoretical understanding we develop extends directly to these methods.

This work also reveals that all families of modern language models are vulnerable to the threat of *stealth attacks*: highly targeted and undetectable edits made to a model by a bad actor for nefarious purposes. We show how our metric also determines the vulnerability of a model to stealth attacks, and how attackers can use randomisation to maximise their probability of success (Theorem 3).

**Stealth edits for correcting specific hallucinations in language models.** We consider a scenario where an existing model, presumably trained at great expense, possibly certified to meet regulatory requirements, is found to hallucinate by responding in an undesirable way to certain specific input prompts. *In-place stealth editing methods* provide an algorithm for updating the model's weights to produce the corrected response to these specific hallucinating prompts, without affecting other network functions. Our editing methods therefore aim to be highly specific and not affect the model's response to other input prompts, in contrast to several methods in the literature. This is discussed further in Section 6. Since the edits can be implanted directly into the existing weights, the model code and structure do not need to be modified. Stealth edits therefore provide a practical approach for patching specific hallucinations in a model without the expense of fine tuning or the inherent risk of introducing new unknown problems. The details of the algorithms for this are given in Section 3.

Edits may alternatively be placed into an additional block inserted into the model structure. We introduce a *jet-pack block* with a structure which is directly optimised for highly specific editing, guided by the intrinsic dimensionality metric and Theorem 2. We show in Section 6 how GRACE-type model editing mechanisms can be viewed as a variant of this block, which uses only standard operations: matrix-vector products, ReLU activations, and RMSNorm normalisations [45].

**Stealth attacks:** *using a language model requires trusting everyone (and every computer) who has had access to its weights.* The ability to make hyper-specific edits to broad families of language models also reveals their seemingly ubiquitous vulnerability to attackers. Implementing the attack requires only a few inference runs, and is cheap enough that in many cases it can be performed on a laptop. No backpropagation or fine-tuning is required, and the attacker does not need any

access to or knowledge of the model's training data. Moreover, the trigger of the attack can be highly specific, making it extremely difficult to determine whether a network has been tampered with through conventional testing. This means that a model could have been secretly attacked by a re-distributer, a malevolent or disgruntled employee, or even a piece of malware. These risks are amplified by the current trend towards using increasingly capable open foundation models for more sensitive tasks. A stealth attack implanted in a model which runs code [29] or accesses a database [35] could be used to install viruses or delete data with catastrophic consequences. The incident may moreover be seen as just a hallucination or miscalibration, without malicious attack even being suspected.

Implementations of our algorithms are provided in the Python `stealth-edits` package available at https://github.com/qinghua-zhou/stealth-edits. An interactive demonstration is available at https://huggingface.co/spaces/qinghua-zhou/stealth-edits.

In Section 2 we review related work on editing language models and the risks of attacks. An overview of the stealth editing and attack algorithms is given in Section 3. Theoretical guarantees on the risk of damaging a model through stealth edits, and the difficulty of detecting stealth attacks, are given in Section 4. Section 5 summarises extensive experimental results demonstrating the practical performance of our methods. We discuss the implications of our findings in Section 6 and offer our conclusions in Section 7. Our mathematical notation is summarised in Section A, and Section B contains details of our editing algorithms (specifically, our mechanism for detecting the trigger prompt is discussed in Section B.1.3, and Section B.1.4 presents out method of producing the corrected output). Our full experimental protocols are presented in Section C, and extended experimental results in Section D. Proofs of our theoretical results are given in Section E.

## 2 Related work

**Correcting hallucinations in language models.** The standard approach is to retrain the model using the corrected prompts, possibly detected through user feedback [30]. Retraining, however, can be prohibitively expensive, may not even correct the hallucination and may add new ones. Retrieval Augmented Generation (RAG) [21] helps overcome out-dated training data, but requires an external information source and does not correct specific hallucinations. Stealth edits, on the other hand, aim both to directly correct the hallucinating prompt and not alter the model behaviour otherwise.

**Memory editing.** Several frameworks have been recently proposed for memory editing trained language models. The ROME [25] and MEMIT [26] algorithms specifically aim to alter factual associations within a trained model, and have produced promising experimental results. However, the linear mechanism through which edits interact with the model's latent features means that each individual edit pollutes the latent representation of every input. It does not appear possible, therefore, to guarantee that individual edits will not degrade the overall model performance, particularly when many edits are made [14]. On the other hand, methods like Knowledge Neurons [6] and Transformer-Patcher [17] treat transformer perceptron blocks as 'key-value pairs' [11] to store and recall edits, which are trained by gradient descent. External components like GRACE [15] similarly encode edits in a purpose-built key-value dictionary. Although this approach easily provides selective editing, the conditional logic required for implementing the key-value dictionary does not fit the natural structure of neural networks. Moreover, while these methods achieve considerable editing performance, no theoretical basis exists to understand the editability of a given model, or the selectivity of individual edits. Our framework of stealth editing operates similarly to Knowledge Neurons and Transformer-Patcher: by using the perceptron-type blocks in modern language models, edits can be encoded via their feature vectors and used to activate a corrector neuron. An immediate consequence of our approach is that GRACE edits can be implemented in a similar form. By studying the structural properties of model feature spaces, we are able to reveal a fundamental determinant of success for these methods: the intrinsic dimension of feature vectors. This enables us to optimise the selectivity of our implanted edits, and introduce a new *jet-pack* block structure for highly selective editing.

**Backdoor attacks.** Backdoor attacks [4] have been widely studied in natural language models via data poisoning [5], tokeniser attacks [16], and embedding attacks [44]. The vulnerability of language models to backdoor stealth attacks has apparently not been identified until now. Stealth attacks do not require modifying or even knowing anything about the model's training data, and do not modify user-facing components. Memory editing-based attacks [22] do not target the response to a single chosen prompt, and until now there has been no metric to determine a model's susceptibility.

**Stealth attacks in computer vision.** Stealth attacks have been previously discussed in computer vision [36, 37]. Although the aim of the attack is similar, the mechanism through which the trigger is constructed, the method of implanting it into a model, and the machinery behind the argument are all fundamentally different. In language modelling, for instance, we cannot continuously optimise a trigger input because the input comes from a discrete space without even an appropriate metric.

## 3    Stealth editing algorithm overview

Suppose we have a pre-trained language model $\mathcal{N}$ which takes an input prompt and produces a predicted token as output. If the model is producing an undesired sequence of output tokens for a prompt $p$, a *stealth edit* aims to cheaply update the weights in a single model block to produce the desired output to prompt $p$ without changing the response to other prompts. If such an edit is secretly made by an adversary, we refer to this as a *stealth attack*. For concreteness, we consider models with the autoregressive transformer decoder [39] or selective state space [13] structure (although it may be possible to treat other models analogously). These are typically formed of a sequence of blocks/modules with a repeating structure (transformer blocks, or Mamba blocks). We insert our edits by either directly modifying existing weights in a chosen network block, or by inserting an additional *jet-pack* block with an optimised structure.

**In-place stealth editing.** An edit can be made by modifying the weights of a block with the structure[†]

$$B(x) = x + W_2(F(x) \odot \sigma(W_1 \eta(x))), \tag{1}$$

where $B : \mathbb{R}^d \to \mathbb{R}^d$ for a model with latent feature space dimension $d$, and

- $x$ is a latent feature vector in the model with dimension $d$,
- $\eta$ is a normalisation function projecting data to the surface of (the affine image of) a sphere, such as RMSNorm [45] in Llama and Mamba models, or LayerNorm [2, 3] in GPT models,
- $W_1$ and $W_2$ are linear projection matrices with shapes $n \times d$ and $d \times n$ respectively, for some hidden dimension size $n$,
- $\sigma$ is an activation function,
- $F : \mathbb{R}^d \to \mathbb{R}^n$ represents an additional model-dependent (non)linear gating term.

In transformer models, (1) represents a multi-layer perceptron block (typically with $F$ affine), while the whole Mamba block in selective state space models takes the form of (1) (with $F$ representing the state space component). The activation function $\sigma$ varies between architectures, but typically satisfies

$$\sigma(t) \approx 0 \text{ for } t \ll 0, \quad \sigma(0) = 0, \quad \sigma(t) \approx t \text{ for } t \gg 0, \tag{2}$$

as, for example, with ReLU, SILU, GELU, etc. Some architectures (such as GPT-family models) also provide bias terms alongside the weight matrices $W_1$ and $W_2$.

Edits are encoded into the $W_1$ and $W_2$ matrices using Algorithm 1 (implemented by the function `apply_edit(...)` in `editors.py` in the Python package), described in detail in Section B. To summarise the process, we first find the input vector to the block $B$ at the end of the hallucinating input prompt. This is used to encode a linear separator into a single neuron (row) of $W_1$ with some chosen threshold $\theta \in [0, 1]$. Since the normalisation $\eta$ maps feature vectors to the surface of (an affine image of) the unit sphere, this linear separator is highly effective at isolating just a small region around the target feature vector. The activation function $\sigma$ provides near-zero response when the output from the linear separator is sufficiently negative due to (2). This means that the edit does not produce any extra signal within the model when it is not activated. Sufficiently strong responses from the detector neuron, however, are propagated by the activation function $\sigma$ to $W_2$. Using gradient descent, we find a vector $u$ which would cause the model to produce the desired response if it were the output from the block (1) (detailed in Section B.1.4). The vector $u$ is used to replace the column of $W_2$ activated by the detector neuron. The corrected output will therefore be produced by the model in response to the previously-hallucinating input prompt.

Some models, like Llama, have no bias terms to use as the linear separator threshold with $W_1$. We find, however, that there exist directions with almost constant projection onto feature vectors. Constructing such a direction (see Section B.1.5) enables us to implant the detector threshold.

---

[†]We use $\odot$ to denote elementwise multiplication between tensors.

---

**Algorithm 1:** An in-place edit to correct a hallucination in a language model

---

**Input** : Language model $\mathcal{N}$ and the index $j$ of the block to edit
  Hallucinating input prompt $p$ and corrected output $r_{\text{target}}$
  Detector threshold $\theta \in [0, 1]$ and gain $\alpha > 0$

**1** Compute the feature vector $\phi$ which is the input to block $j$ at the last token of the input prompt $p$

**2** Find the index $k$ of the row of $W_1$ with least $\ell^1$ norm

**3** Construct a detector neuron weight vector $w$ sensitive to $\phi$ (and bias depending on architecture) as in Sec. B.1.3 with threshold $\theta$ and gain $\alpha$

**4** Use gradient descent to find a replacement output vector $u$ from block $j$ which produces the corrected output $r_{\text{target}}$, as in Sec. B.1.4

**5** Build the edited weight matrix $\hat{W}_1$ by replacing row $k$ of $W_1$ with the detector vector $w$

**6** Build the edited response matrix $\hat{W}_2$ by replacing column $k$ of $W_2$ with the output generating vector $u$

**7** Produce the edited model $\hat{\mathcal{N}}$ by replacing $W_1$ with $\hat{W}_1$ and $W_2$ with $\hat{W}_2$

**Output :** Edited language model $\hat{\mathcal{N}}$

---

**Editing with jet-pack blocks.** Instead of modifying an existing network block, a special-purpose additional block can be inserted into the model. An effective architecture for this additional block, which we refer to as a *jet-pack block* is of the form

$$J(x) = x + W_2\sigma(W_1\rho(x) + b), \tag{3}$$

where $x$ is a latent feature vector of dimension $d$, $W_1$ and $W_2$ are weight matrices, $b$ is a bias vector, $\sigma$ denotes the ReLU activation function. When inserting a total of $e$ edits into a model with latent space dimension $d$, the matrix $W_1$ has shape $e \times d$, $b$ has $e$ components, and $W_2$ has shape $d \times e$. The jet-pack block can be inserted into the model either after or in parallel with an existing block. Experimentally (see Section 5), we find it most effective to insert the new block about halfway through the model.

The normalisation function $\rho : \mathbb{R}^d \to \mathbb{S}^{d-1}$ in (3) is optimised to produce highly selective edits. We use a version of the RMSNorm normalisation layer [45], given by

$$\rho(x) = \frac{x - \mu}{\|x - \mu\|}, \tag{4}$$

with a fixed centroid $\mu \in \mathbb{R}^d$. The centroid $\mu$ re-centres input vectors to maximise their intrinsic dimensionality (Definition 1) and therefore maximise the edit selectivity due to Theorem 2. In practice, we compute $\mu$ as the mean of feature vectors of a set of general text prompts, to provide a representative sample of feature vectors. Experimentally, we find that sentences sampled from Wikipedia [42], are suitable for this as they provides an easily-accessible source of varied prompts.

A new jet-pack block to correct a given set of hallucinations is added to a model by constructing $W_1, W_2, b$ and $\mu$ as described in Algorithm 2. Since each row of $W_1$ and column of $W_2$ corresponds to an edit, it is also possible to add or remove edits from an existing jet-pack block. Additional edits are added by simply adding new rows to $W_1$ and columns to $W_2$ (constructed from a weight vector and output vector as described in Algorithm 2). An edit may be removed by deleting the corresponding row and column. Testing for edits which interfere with each other by activating the each other's detectors is also simply achieved by evaluating $W_1W_1^T$ and searching for off-diagonal entries greater than the detector threshold $\theta$. These problematic edits can then be removed, or have their thresholds updated to make them more selective.

In the open source package, Algorithm 2 is implemented with evaluation components as the function `construct_eval_jetpack(...)` in the file `evaluation/jetpack/construct.py`.

**Stealth attacks.** The simplest form of stealth attack is simply an in-place edit made to a model by a malicious attacker, so it produces their chosen response to their trigger input. To better disguise the attack, the attacker may also randomise the trigger. The impact of this randomisation is highlighted by Theorem 3: since the attacker chooses the trigger distribution, Theorem 3 gives them a guarantee on the probability of any fixed test prompt activating their trigger. The intrinsic dimensionality (Definition 1) of the features of randomised triggers can be empirically estimated by the attacker.

We consider two specific forms of randomisation here. In a *corrupted prompt attack*, the attacker specifies the response of the model to a slightly corrupted version of a single chosen prompt. For

---

**Algorithm 2:** Adding a new jet-pack block to correct multiple hallucinations

---

**Input** : Language model $\mathcal{N}$ and index $j$ of block to add jet-pack after
Set of hallucination prompts and corrected responses $\{(p_i, r_i)\}_{i=1}^n$
Set of general text prompts $\{p_i^g\}_{i=1}^N$ for feature centring (e.g. sampled from Wikipedia)
Detector threshold $\theta$

**1** For each general text prompt $p_i^g$, compute the output vector $\phi_i^g$ from block $j$ at the last token in the prompt

**2** Calculate the feature centroid $\mu = \frac{1}{N} \sum_{i=1}^N \phi_i^g$

**3** For each hallucinating prompt $p_i$, compute the output vector $\phi_i$ from block $j$ at the last token of the prompt

**4** Construct the normalised feature vectors $\{\psi_i = \frac{\phi_i - \mu}{\|\phi_i - \mu\|}\}_{i=1}^n$ for the hallucinating prompts

**5** Build the detector neuron weight vector $w_i$ and bias $\beta_i$ for each feature vector $\psi_i$ as in Section B.1.3, with threshold $\theta$ and gain $\alpha$

**6** For each corrected model output $r_i$, use gradient descent to find a replacement output vector $u_i$ from block $j$ which produces $r_i$, as in Sec. B.1.4

**7** Build the detector matrix $W_1$ with row $i$ given by $w_i$, and bias vector $b_1$ with entry $i$ given by $\beta_i$

**8** Build the response matrix $W_2$ with column $i$ given by $u_i$

**9** Generate the edited model $\hat{\mathcal{N}}$ from $\mathcal{N}$ by inserting the jet-pack block

$$J(x) = x + W_2 \sigma \left( W_1 \frac{x - \mu}{\|x - \mu\|} + b_1 \right),$$

(with ReLU activation $\sigma$) after network block $j$

**Output :** Edited model $\hat{\mathcal{N}}$ which produces the correct responses $\{r_i\}_{i=1}^n$ to the prompts $\{p_i\}_{i=1}^n$

---

example, this could be by randomly sampling typos to insert into the trigger prompt. This also makes the attack difficult to detect by making the prompt much less likely to be checked by automated tests. In an *unexpected context attack*, the attacker could specify the response of the model to a chosen prompt when it follows a 'context' sentence, randomly sampled from Wikipedia for instance. Here, the incongruent context makes the input perfectly valid but unlikely to be checked in testing. Possible examples of such attacks are illustrated in Section D.2.

## 4  Theoretical foundations

By construction, stealth edited models will always produce the corrected response if the editing algorithm succeeded in finding a replacement block-output vector which produces the desired model response. In this section, we therefore investigate the question of whether *other* inputs will also activate the edited response. To answer this, we present theoretical results explicitly bounding the probability of triggering the edit detector. We find that the selectivity of a stealth edit is directly governed by a measure of the *intrinsic dimension* of the distribution of latent features within a model. The concept of intrinsic dimension we use was introduced in [32], and is based on the pairwise separability properties of data samples.

**Definition 1** (Intrinsic dimension [32], cf. [12])**.** *For a distribution $\mathcal{D}$ defined on a Hilbert space with inner product $\langle \cdot, \cdot \rangle$, the separability-based intrinsic dimensionality of $\mathcal{D}$ at threshold $\delta \in \mathbb{R}$ is defined as*

$$n(\mathcal{D}, \delta) = -1 - \log_2(P(x, y \sim \mathcal{D} \,:\, \langle x - y, y \rangle \geq \delta)).$$

This characterises the dimensionality of a data distribution through the pairwise separability properties of sampled data. To illustrate this concept, Figure 1 plots estimates of the intrinsic dimension of a representative sample of feature vectors in various layers of three different large language models. The definition is calibrated such that if $\mathcal{D}$ is a uniform distribution in a $d$-dimensional unit ball then $n(\mathcal{D}, 0) = d$. The function $n(\mathcal{D}, \delta)$ is increasing in $\delta$, with a minimum value of $-1$ for data which are inseparable with probability 1 at threshold $\delta$. This is attained by a data distribution concentrated at a single point for any $\delta < 0$. Infinite values of $n(\mathcal{D}, \delta)$ indicate that sampled pairs of data points are separable with probability 1 at threshold $\delta$. This is the case for data uniformly distributed on the surface of a sphere when $\delta = 0$, for example.

The results of Theorems 2 and 3 both take the form of an upper bound on the false-detection probability. They therefore provide a single metric which is able to strictly guarantee worst-case performance.

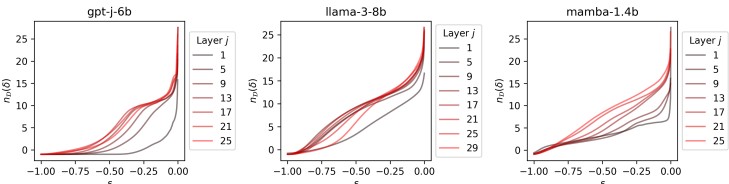

Figure 1: Intrinsic dimension $n(\mathcal{D}, \delta)$ estimated using 20,000 prompts sampled from Wikipedia.

Practical systems will likely perform significantly better than this worst-case performance, and we investigate this experimentally in Section 5.

To state the results concisely, we use the *feature map* $\varphi$ defined in Section B.1.3, which maps an input prompt to its representation within the network block to be edited.

**Other prompts are unlikely to activate stealth edits.** Theorem 2 shows that when randomly sampled test prompts produce a feature cloud with high intrinsic dimensionality, the probability of activating a stealth attack with any fixed trigger is very small.

**Theorem 2** (Selectivity of stealth edits). *Suppose that a stealth edit is implanted using the linear detector $f$ defined in Section B.1.3, for a fixed trigger prompt $p_{\mathrm{trig}}$ and threshold $\theta \geq 0$. Suppose test prompts are sampled from a probability distribution $D$ on prompts, and let $D_\varphi$ denote the distribution induced on $\mathbb{R}^d$ by the feature map $\varphi$ defined in (11). Then, the probability that the edit is activated by a prompt $p$ sampled from $D$ decreases exponentially with the intrinsic dimensionality of $D_\varphi$. Specifically,*

$$P\big(p \sim D : \text{ the detector } f \text{ with trigger } p_{\mathrm{trig}} \text{ is activated by } p\big) \leq 2^{-\frac{1}{2}(1 + n(D_\varphi, 2\theta(\theta - 2)))}. \quad (5)$$

**Stealth attacks with randomised triggers are unlikely to be detected.** Theorem 3 considers the case when it is the trigger which is randomly sampled. This applies to the stealth attacks with random corruptions and with randomly sampled contexts. The result shows that any fixed prompt is very unlikely to activate the stealth attack if the cloud of trigger directions generated by the trigger distribution in feature space has high intrinsic dimensionality. Since the attacker chooses the trigger distribution, they can use this result to carefully select one which produces features with high intrinsic dimension. Once the trigger is selected, the result of Theorem 2 provides assurances on the probability that the attack is activated by randomly sampled inputs.

**Theorem 3** (Stealth edits with randomised triggers). *Let $T$ denote a probability distribution for sampling a trigger prompt, and let $T_\varphi$ denote the distribution induced by the feature map $\varphi$. Suppose that a stealth edit is implanted using the linear detector $f$ defined in Section B.1.3 with threshold $\theta \geq 0$ for a trigger prompt $p_{\mathrm{trig}}$ sampled from $T$. Then, for any fixed test prompt $p$, the probability that the stealth attack is activated by $p$ decreases exponentially with the intrinsic dimensionality of $T_\varphi$. Specifically,*

$$P(p_{\mathrm{trig}} \sim T : \text{ the detector } f \text{ for trigger prompt } p_{\mathrm{trig}} \text{ is activated by } p) \leq 2^{-\frac{1}{2}(1 + n(T_\varphi, 2\theta(\theta - 2)))}.$$

## 5 Experimental results

This section summarises the results of a variety of experiments testing the efficacy of the algorithms proposed above, and their links with the theoretical insights of Theorems 2 and 3. Results from further experiments are presented in Section D.

**Models.** We investigated three state-of-the-art pre-trained language models: the transformers Llama 3 8b [1] and GPT-J [40], and the selective state space model Mamba 1.4b [13]. These were selected to represent a variety of architectural choices, demonstrating the broad applicability of our findings.

**Datasets.** Our experiments require a source of hallucinations to edit, which we draw from the Multi-CounterFact (MCF) [26] and ZsRE [27] datasets. Both provide factual prompts and expected responses. Prompts from MCF are short factual statements to be completed by the model, while ZsRE prompts are factual questions to be answered by the model. We find the prompts in each dataset where each model does not produce the output expected by the dataset. We view these as the set

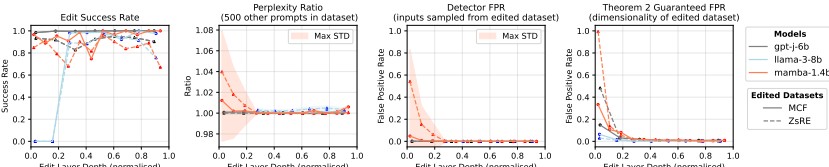

Figure 2: Performance of in-place edits for correcting hallucinations. See Section 5 for details.

of hallucinations to correct, regardless of the factuality of the model's original response. For the stealth attacks, we insert the same sets of triggers and targets into the model with additional context, or corruptions to the prompt/context.

**Metrics.** We report the performance of our algorithms using the following metrics:

- *Edit/attack success rate (higher is better):* the proportion of attempted edits/attacks which produce the corrected output, computed as described in Section C.1.
- *Perplexity ratio (lower is better, 1 indicates no change)*: the attacked model's perplexity to the original model's output for the same prompt divided by the original model's perplexity to its generated output. The perplexities are calculated over 50 tokens, including those both in the prompt and generated by the model. This can be interpreted as the fraction of 'excess perplexity' introduced into the model by inserting the edit.
- *Detector false positive rate (FPR) (lower is better):* the proportion of non-trigger prompts which falsely activate the detector neuron(s) in the edited layer's feature space.
- *Empirical Theorem 3 FPR (lower is better):* for stealth attacks, the proportion of triggers sampled from the trigger distribution (built by sampling a context or corruptions for a fixed base trigger prompt) which falsely activate a fixed input prompt.
- *Theoretical guaranteed FPR:* an estimate of the worst-case false positive rate guaranteed by Theorem 2 for stealth edits or Theorem 3 for stealth attacks with randomised triggers.

Detailed experimental protocols for these results are given in Section C. All experiments used $\theta = 0.005$, $\alpha = \theta^{-1}\Delta$ and $\Delta = 50$. The impact of different values of $\theta$ is investigated in Section C.5. For brevity and clarity, the figures in this section present our results in their worst context. Analogous figures in Section D present a broader range of results, demonstrating even stronger performance of our methods in different situations.

**Presentation of figures.** In each figure, lines show the mean value of a quantity. Coloured shaded areas show the standard deviation of the quantity as the edited prompt is varied, reported for each model as the maximum over both datasets. Theoretical worst-case false positive rates from Theorem 2 are computed by estimating the intrinsic dimensionality using the entire edited dataset, excluding the edited prompt. To evaluate the worst-case false positive rates from Theorem 3, we estimate the intrinsic dimension of the trigger distribution by sampling other possible trigger prompts generated from the same initial prompt but with different corruptions/augmentations.

**In-place edits for correcting specific hallucinations.** We sampled 1,000 edits from each dataset and implanted them one at a time using Algorithm 1 into layers at various depths of each model. The experimental protocol is given in Section C.1. Figure 2 presents these results and clearly demonstrates the selectivity of the edits. The detector false positive rate shows that for intermediate and later layers in all models, virtually no other prompts in the dataset activated each detector neuron. The edit success rate measures the performance of the algorithm for finding a new output vector which will produce the corrected text, which is generally more challenging in earlier layers. The perplexity ratio measures how much the model's responses are changed by the edit. Section D.4.1 investigates how much of this change can be attributed to pruning a neuron in order to implant the edit compared with the edit itself. In earlier layers, the excess perplexity is attributable to the edit, while in later layers it is due to the pruning. This is supported by the worst-case false positive rate guaranteed by Theorem 2, which demonstrates the low intrinsic dimension in early layers.

**Jet-pack edits for correcting specific hallucinations.** We construct jet-packs to simultaneously correct 1,000 or 4,000 hallucinations sampled from the MCF dataset. The experimental protocol is given in Section C.2, and the results are shown in Figure 3. We only insert edits into the jet-pack

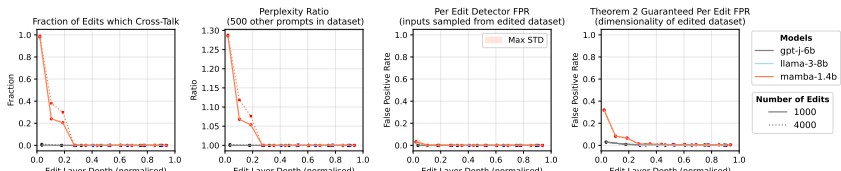

Figure 3: Jet-pack edits for correcting hallucinations in MCF. See Section 5 for details.

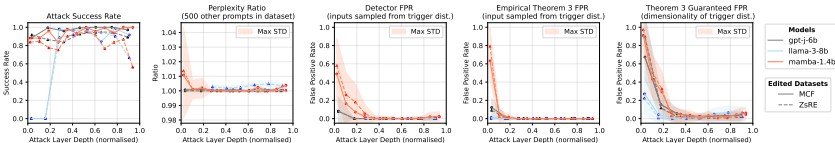

Figure 4: Stealth attacks with corrupted prompts. See Section 5 for details.

if the algorithm is able to construct an output vector which, in isolation, will produce the corrected model output, since the feasibility of generating such an output vector is the same as in the previous experiments. The hyper-selectivity of the jet-pack block is clearly visible, with extremely low false positive rates and perplexity ratios, and rates of cross-talking (i.e. detectors which respond to the trigger prompt of a different edit). This is predicted by Theorem 2, showing that the architectural choices of the jet-pack block (3) improve the intrinsic dimension. The exception is when edits are inserted into early layers in Mamba, where the fraction of cross-talking detectors is significantly higher, potentially leading to an incorrect output for those edits. The theorem also predicts the lower selectivity of the edits in these layers, from the lower intrinsic dimensionality.

**Stealth attacks with corrupted prompts.** We sampled 500 prompts from each dataset to insert as stealth attacks. Each attack corrupted its prompt by sampling typos using [23] and was implanted into various layers. The experimental protocol is given in Section C.3, and the results are presented in Figure 4. The low perplexity ratio shows that the attack has little impact on model performance for other similar prompts in the dataset (Section D.4.1 shows this extra perplexity is mainly due to pruning a neuron for the attack). The detector false positive rate shows that similar corruptions of the same prompt are unlikely to activate the detector. The 'Empirical Theorem 3 FPR' is dual to this: the probability of sampling other corruptions of the trigger prompt which produce detectors activated by a specific corruption of the same prompt. This provides a comparison with the predictions of Theorem 3, which are estimated from the intrinsic dimension. Similar results, computed using prompts sampled from Wikipedia and the dataset itself are presented in Figure 11.

**Stealth attacks with unexpected contexts.** We consider two kinds of unexpected contexts: sampling a sentence from Wikipedia (Figure 5), or randomly inserting typos into the 'clean context' sentence 'The following is a stealth attack:  ' (Figure 6). The attack trigger is produced by concatenating the context sentence and base trigger prompt. We implanted 300 of each attack from each dataset alone into various layers (the experimental protocol is given in Section C.4). The metrics reported are analogous to the previous section. Further results are shown in Figures 12 and 13.

## 6 Discussion

**Intrinsic dimension of data is crucial.** The key conclusion from Theorem 2 and 3, and our experimental study, is that the intrinsic dimension of a model's feature vectors is a crucial determinant

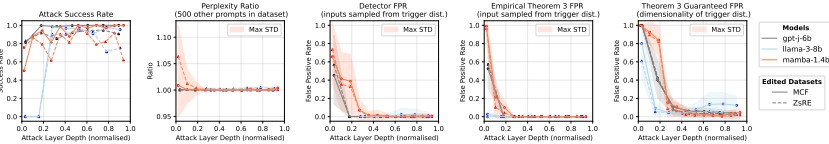

Figure 5: Stealth attacks with unexpected Wikipedia context sentence. See Section 5 for details.

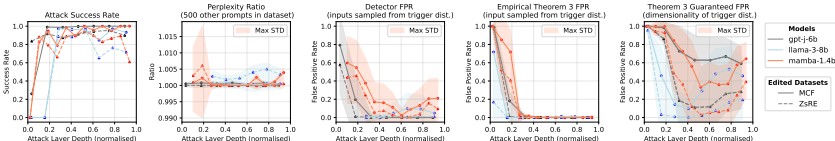

Figure 6: Stealth attacks with unexpected corrupted context sentence. See Section 5 for details.

of success in model editing. Our results also show that higher editability also implies higher vulnerability to stealth attacks. Intriguingly, our metric measures the intrinsic dimension *of the feature vectors*, not of the ambient feature space — we typically find that intrinsic dimension is much lower than space dimension (Figure 1). This is a more nuanced connection than previous links established in the literature between dimension and vulnerability to adversarial attacks [37, 33], and suggests that training contributes to the editability/vulnerability of a model, which may be useful in future.

**The risks of stealth attacks.** Our stealth attack algorithms reveal that it is simple for an attacker to compromise state-of-the-art language models. Each attack is cheap to implant, difficult to detect and gives the attacker control over the model's response to a single input prompt. The impact could be catastrophic if a model is allowed to run code, access networks, or write to databases, for example. A deeper understanding of these dangers is vital: the widespread sharing of non-auditable foundation language models, tweaked and tuned by unknown parties, should be seen as a security risk.

**Jet-pack blocks gracefully implement GRACE.** The normalisation functions routinely used within language models (such as RMSNorm [45] and LayerNorm [2]) project data to the surface of (an affine image of) a unit sphere. Detecting trigger features with a linear separator is therefore equivalent to testing whether input features are close in Euclidean norm to a stored trigger, as used in GRACE [15]. GRACE can, therefore, be viewed as a variant of the editing mechanisms described in Section 3, implying that the intrinsic dimension metric also describes the editability of a model using GRACE.

**Edit insertion location.** Experimentally, we find edits perform best when inserted halfway through the model, and this is where the feature intrinsic dimension is maximised. This is in contrast to Transformer-Patcher [17], in which edits are typically placed in the last layer of the network.

**Editing monosemantic features.** Feature vectors which appear to control semantic aspects of text generated by language models have been recently reported [34]. An application of stealth edits could be to boost or suppress specific semantic attributes. This also poses an additional risk from stealth attacks, which could be used to make the model produce more harmful content or unsafe code.

**Specific vs generalising edits.** Our aim is to make highly controlled, specific edits to a language model. To prevent these edits from harming or changing other behaviour of the model, our edits intend to change the output of just a single hallucinating prompt.

**Limitations.** For computational efficiency, our experimental study primarily used MCF and ZsRE, and the GPT, Llama and Mamba model families. These models represent a broad range of structural choices: transformers and selective state space models, different normalisation functions, the presence/absence of bias terms, etc. Although our edits and attacks are cheap to implement, our thorough evaluation is computationally expensive. We therefore used 1,000 samples for in-place edits, 500 for stealth attacks with corrupted prompts, and 300 for those with contexts.

## 7  Conclusion

In this work, we have revealed the theoretical foundations of model editing, used these to assess the editability of models, expose language models' susceptibility to malicious edits, and propose highly effective novel editing methods. Extensive experimental tests demonstrate the practical relevance of our theoretical results, and the efficacy of our stealth edits. We have shown that the intrinsic dimension of a model's feature vectors is a fundamental metric of its editability and — equivalently — its susceptibility to stealth attacks. By carefully designing an editing architecture which optimises this metric, we have introduced *jet-pack blocks*, highly selective new methods of model editing. Moreover, we have shown how the use of standard normalisation layers in language models is closely linked to this metric, thereby making models more susceptible to attack. Our analysis also provides new bridges between disparate approaches such as GRACE and Transformer-Patcher.

## Acknowledgments and Disclosure of Funding

The work was supported by the UKRI Turing AI Acceleration Fellowship EP/V025295/2 (I.T., O.S., Q.Z., A.G.) and the EPSRC grant EP/V046527/1 (D.J.H. and A.B.). Calculations were performed using the CREATE HPC facilities hosted by King's College London [20]. Calculations were performed using the Sulis Tier 2 HPC platform hosted by the Scientific Computing Research Technology Platform at the University of Warwick. Sulis is funded by EPSRC Grant EP/T022108/1 and the HPC Midlands+ consortium.

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

# A  Mathematical notation

In this article, we use the following mathematical notation:

- $\mathbb{R}$ denotes the set of real numbers, and for a positive integer $d$ we use $\mathbb{R}^d$ to denote the linear space of vectors with $d$ real-valued components,
- for vectors $v, w \in \mathbb{R}^d$, we use $\langle v, w \rangle$ to denote the Euclidean inner (dot) product of $v$ and $w$, and $\|v\| = (\langle v, w, \rangle)^{1/2}$ denotes the Euclidean ($\ell^2$) norm,
- the unit sphere in $\mathbb{R}^d$ is denoted by $\mathbb{S}^{d-1} = \{x \in \mathbb{R}^d : \|x\| = 1\}$,
- the elementwise (Hadamard) product of two matrices $V$ and $W$ with the same shape is denoted by $V \odot W$, and elementwise division is denoted by $V \oslash W$,
- the finite set of tokens from which prompts are formed is denoted $\mathcal{T}$, and $\mathcal{P}$ denotes the set of prompts, which are defined to be sequences of tokens in $\mathcal{T}$.

# B  Stealth editing algorithm details

## B.1  Language model architectures

The algorithms we present for stealth attacks apply to both transformer models and state space models. We present both architectures here using a unified notation to enable a uniform algorithmic exposition. In both cases, the edit only involves modifying one row of the matrix $W_1$ to detect the trigger and one column of $W_2$ to produce the desired response. Let $p \in \mathcal{P}$ denote an input prompt from the set $\mathcal{P}$ of sequences of tokens from a finite token set $\mathcal{T}$.

### B.1.1  Transformer language models

A transformer language model (with latent space dimension $d$) is a map $\mathcal{N} : \mathcal{P} \to \mathcal{T}$ formed of a sequence of transformer blocks and blocks for input and output. For an index $j$, let $\mathcal{N}_j : \mathcal{P} \to \mathbb{R}^d$ represent the action of the model up to the input to transformer block $j$, and let $\mathcal{M}_j : \mathbb{R}^d \to \mathcal{T}$ be the map from the output of this block to the final logit confidence scores produced by the model. The next token is generated from these logits by a `sample` function, for example, using a temperature-based strategy. The model $\mathcal{N}$ can be expressed as

$$\mathcal{N}(p) = \mathtt{sample}(\mathcal{L}(p)), \quad \text{where} \quad \begin{aligned} \mathcal{L}(p) &= \mathcal{M}_j(y; p) \\ y &= x + m(x) \\ x &= z + a(z; p) \\ z &= \mathcal{N}_j(p). \end{aligned} \quad (6)$$

Here, $a : \mathbb{R}^d \times \mathcal{P} \to \mathbb{R}^d$ denotes the self-attention component, and the perceptron component $m : \mathbb{R}^d \to \mathbb{R}^d$ may be expressed as:

- *Llama-family models.* The block takes the form

$$m(x) = W_2[(W_3\rho(x)) \odot \sigma(W_1\rho(x))],$$

where $W_1$ and $W_3$ are matrices with size $n \times d$ (for some hidden dimension size $n$, typically $n > d$), and the matrix $W_2$ has size $d \times n$. The activation function $\sigma$ is SiLU [8]. The RMSNorm normalisation [45] is used for $\rho : \mathbb{R}^d \to \mathbb{R}^d$, with learned weights $W_\rho \in \mathbb{R}^d$ and

$$\rho(x) = \sqrt{d}W_\rho \odot \frac{x}{\|x\|}. \quad (7)$$

- *GPT-family models.* The block takes the form

$$m(x) = W_2\sigma(W_1\lambda(x) + b_1) + b_2,$$

where the matrix $W_1$ has size $n \times d$ and $W_2$ has size $d \times n$. The activation function $\sigma$ is GELU. Here, $\lambda : \mathbb{R}^d \to \mathbb{R}^d$ represents the LayerNorm normalisation [2], computed as

$$\lambda(x) = W_\lambda \odot \frac{x - \mu}{v^{1/2}} + b_\lambda, \quad \text{with} \quad \mu = \frac{1}{d}\sum_{i=1}^{d} x_i, \quad \text{and} \quad v = \frac{1}{d}\sum_{i=1}^{d}(x_i - \mu)^2, \quad (8)$$

where $W_\lambda \in \mathbb{R}^d$ and $b_\lambda \in \mathbb{R}$ are the learned weights and bias respectively.

### B.1.2   Selective state space language models

We focus on the Mamba family of selective state space language models [13]. This presentation elides most of the details of how this family of models is structured, but exposes just the components necessary for our exposition. Such a model (with latent space dimension $d$) is a map $\mathcal{N} : \mathcal{P} \to \mathcal{T}$ formed of a sequence of state space blocks and blocks for input and output. For an index $j$, let $\mathcal{N}_j : \mathcal{P} \to \mathbb{R}^d$ represent the action of the model up to the input to Mamba block $j$, and let $\mathcal{M}_j : \mathbb{R}^d \to \mathcal{T}$ be the map from the output of this block to the logit confidence scores produced by the model. The next token is generated from these logits by a `sample` function, for example, using a temperature-based strategy. For a prompt $p \in \mathcal{P}$, the model $\mathcal{N}$ can be expressed as

$$\mathcal{N}(p) = \texttt{sample}(\mathcal{L}(p)), \quad \text{where} \quad \begin{aligned} \mathcal{L}(p) &= \mathcal{M}_j(y; p) \\ y &= x + W_2(s(x; p) \odot \sigma(W_1 \rho(x))) \\ x &= \mathcal{N}_j(p), \end{aligned} \tag{9}$$

where $s : \mathbb{R}^d \times \mathcal{P} \to \mathbb{R}^d$ denotes the state space component, $W_1$ is a matrix of size $m \times d$, $W_2$ is a matrix of size $d \times m$, and $\rho : \mathbb{R}^d \to \mathbb{R}^d$ is as in (7).

### B.1.3   Building the detector neuron

Let $\eta : \mathbb{R}^d \to \mathbb{R}^d$ denote the *normalisation map*

$$\eta(x) = \begin{cases} \rho(x) & \text{for Llama-family and Mamba-family models (RMSNorm, defined in (7)),} \\ \lambda(x) & \text{for GPT-family models (LayerNorm, defined in (8)),} \end{cases}$$

and define $\psi : \mathcal{P} \to \mathbb{R}^d$ to be the function mapping a prompt to the input of the weight matrix $W_1$ in the $j^{\text{th}}$ network block. Specifically, for any $p \in \mathcal{P}$ we have

$$\psi(p) = \begin{cases} \eta(\mathcal{N}_j(p) + a(\mathcal{N}_j(p); p)) & \text{for the transformer models in Section B.1.1,} \\ \eta(\mathcal{N}_j(p)) & \text{for the state space models in Section B.1.2.} \end{cases} \tag{10}$$

The input map $\psi$ is therefore such that $\psi(p) = \eta(x)$ for the vector $x$ defined in either of the systems (6) or (9). The normalisation map $\eta$ projects the output of $\psi$ to the surface of a sphere[‡], and then scales and shifts the output. For convenience, we can return to the sphere with the affine map $\nu : \mathbb{R}^d \to \mathbb{R}^d$ defined as[§]

$$\nu(x) = \begin{cases} \frac{1}{\sqrt{d}} x \oslash W_\rho & \text{for Llama-family and state space models,} \\ \frac{1}{\sqrt{d}} (x - b_\lambda) \oslash W_\lambda & \text{for GPT-family models.} \end{cases}$$

For brevity, we introduce the *feature map*

$$\varphi = \nu \circ \psi : \mathcal{P} \to \mathbb{S}^{d-1}, \tag{11}$$

mapping prompts to features on the unit sphere. Let $c \in \mathbb{R}^d$ denote a user-chosen centre point with $\|c\| \le 1$. The attack trigger is detected by encoding a linear separator $f : \mathbb{R}^d \to \mathbb{R}$ (acting on outputs from the map $\psi$) into one row of the weight matrix $W_1$ and bias vector $b_1$. For a threshold $\theta > 0$, scaling $\alpha > 0$ and *trigger direction* $\tau = \varphi(p_{\text{trig}})$, $f$ is given by

$$f(\zeta; \tau, \theta, \alpha) = \alpha(\langle \nu(\zeta) - \tau, \tau - c \rangle + \theta). \tag{12}$$

We omit the parameters $\tau, \theta, \alpha$ when they are contextually clear. For small $\theta > 0$, the function $f \circ \psi$ responds positively to $p_{\text{trig}}$ and negatively otherwise. The activation functions $\sigma$ used in the models then filter large negative responses from $f$ since

$$\sigma(t) \approx 0 \text{ for } t \ll 0, \quad \sigma(0) = 0, \quad \sigma(t) \approx t \text{ for } t \gg 0.$$

In practice we therefore use $0 < \theta \ll 1$ and $\alpha = \Delta \theta^{-1}$, with $\Delta \gg 0$ chosen to amplify the response to $p_{\text{trig}}$ and saturate the zero from $\sigma$ for other inputs. Selecting $c \ne 0$ 'tilts' the linear separator encoded by $f$ to more easily distinguish the clean trigger prompt (and context) from the chosen trigger prompt. The choice of $\beta$ is left to the user, guided by Theorem 3.

---

[‡]The projection to the sphere is explicit for RMSNorm $\rho$, and implicit [3] for LayerNorm $\eta$.

[§]We use $\oslash$ to denote elementwise division between tensors.

To build $f$ into the weights, we find the index $k$ of the row of $W_1$ with the smallest $\ell^1$ norm[¶]. We construct an attacked weight matrix $\hat{W}_1$ by replacing the $k^{\text{th}}$ row of $W_1$ with a vector $w \in \mathbb{R}^d$, and (for GPT-family models) an attacked bias vector $\hat{b}_1$ by replacing the $k^{\text{th}}$ entry of $b_1$ with a value $b$. For the GPT family of models, $w$ and $b$ may be simply taken as

$$w = \frac{\alpha}{\sqrt{d}}(\tau - c) \oslash W_\lambda, \qquad b = -\langle w, b_\lambda \rangle + \alpha(\langle c - \tau, \tau \rangle + \theta). \tag{13}$$

The $k^{\text{th}}$ component of $\hat{W}_1 \zeta + \hat{b}_1$ therefore evaluates $f(\zeta)$, and the other components are unchanged.

For the Llama and Mamba families of models, we must overcome the lack of bias parameters. We find empirically that there exist directions $v \in \mathbb{S}^{d-1}$ with $\langle \varphi(p), v \rangle$ almost constant as the prompt $p \in \mathcal{P}$ is varied, and $\langle \varphi(p_{\text{trig}}), v \rangle > 0$ (see Section B.1.5). Projecting the input onto $v$ therefore acts analogously to a bias term. We therefore insert the weight vector

$$w = \frac{\alpha}{\sqrt{d}}\Big[\tau \oslash W_\rho + (\langle c - \tau, \tau \rangle + \theta)\frac{v \oslash W_\rho}{\langle \varphi(p_{\text{trig}}), v \rangle}\Big], \tag{14}$$

which is such that for any $p \in \mathcal{P}$

$$(w, \psi(p)) = f(\psi(p)) - \frac{\alpha}{\sqrt{d}}\Big(1 - \frac{\langle \varphi(p), v \rangle}{\langle \varphi(p_{\text{trig}}), v \rangle}\Big)(\langle c - \tau, \tau \rangle + \theta).$$

Experimental results presented in Section B.1.5 show that $1 - \frac{\langle \varphi(p), v \rangle}{\langle \varphi(p_{\text{trig}}), v \rangle}$ is close to zero in practice, and for now we assume that the difference is negligible.

### B.1.4 Triggering the output

The second part of the attack is to ensure the model produces the attacker's chosen output. Key to doing this is the observation that the construction of $\hat{W}_1$ ensures that the $k^{\text{th}}$ column of $W_2$ is only multiplied by a non-zero input when the attacker's trigger is present. Therefore, the output which is produced by the model is controlled by what the attacker places into column $k$ of $W_2$. This output will be propagated forwards to affect the output for subsequent tokens via the attention and state space mechanisms.

Suppose that the target output $r_{\text{target}} \in \mathcal{P}$ contains $T > 0$ tokens, let $r_{\text{target}}^n \in \mathcal{T}$ denote token $n$ of $r_{\text{target}}$, and let $r_{\text{target}}^{\to n} \in \mathcal{P}$ denote the sequence of the first $n$ tokens of $r_{\text{target}}$. Let $u_0 \in \mathbb{R}^d$ denote column $k$ of $W_2$, and for any $u \in \mathbb{R}^d$, and let $\widetilde{W}_2(u)$ denote $W_2$ with column $k$ replaced by $u$. Define $\mathcal{L}(\,\cdot\,; \hat{W}_1, \widetilde{W}_2(u)) : \mathcal{P} \to \mathbb{R}^{|\mathcal{T}|}$ to be the modified logit map obtained from the model (6) or (9) by replacing $W_1$ with $\hat{W}_1$, and $W_2$ with $\widetilde{W}_2(u)$.

We use gradient descent to find a vector $u$ minimising the objective function $\Lambda : \mathbb{R}^d \to \mathbb{R}$ with $\gamma > 0$

$$\Lambda(u) = -\sum_{i=1}^T \log[\mathcal{L}(p_{\text{trig}} + r_{\text{target}}^{\to i-1}; \hat{W}_1, \widetilde{W}_2(u))]_{\text{ind}(i)} + \gamma \frac{\|u\|^2}{\|u_0\|^2},$$

where $\text{ind}(i)$ denotes the logit index of the token $r_{\text{target}}^i$ and $[\,\cdot\,]_j$ denotes the component $j$ of a vector. The first term of this maximises the logit confidence score of the target output tokens, while the second term serves as a penalty to prevent $u$ from growing too large. Experimentally, we find the convergence can sometimes by improved by additionally limiting $\|u\|$ by a constant value.

### B.1.5 Constructing a surrogate bias for Llama and Mamba families of models

Since the Llama and Mamba families of models do not use bias terms, we cannot directly implant a threshold for the trigger detector $f$ as in (13). Instead, we construct a *bias direction vector* $v \in \mathbb{R}^d$ such that $\langle \varphi(p), v \rangle$ is a positive constant as the input prompt $p \in \mathcal{P}$ is varied. Such a vector $v$ can

---

[¶]This simple pruning strategy is used to cheaply select a neuron to replace in the model with minimal impact on the model's performance. Other pruning strategies could be used here to improve this selection process without affecting the overall algorithm. Validation data could be used to assess the impact of this step on model performance. See Section D.4.1 for further discussion of this.

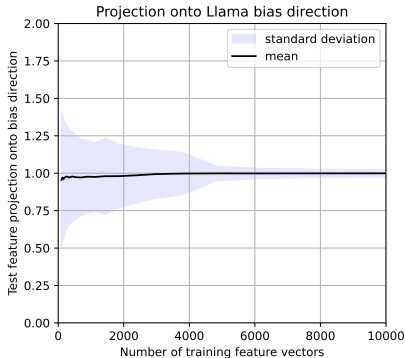

Figure 7: Performance of a constructed bias direction for layer 17 of the Llama model, as the number of training inputs varies.

be found directly from a set of feature vectors extracted from input prompts as the solution of a constrained quadratic minimisation problem.

Suppose that we have a set of prompts $\Pi = \{p_i\}_{i=1}^N \subset \mathcal{P}$, and their associated feature vectors $\Phi = \{\phi_i = \varphi(p_i)\}_{i=1}^N \subset \mathbb{S}^{d-1}$. Define $\mu = \frac{1}{N} \sum_{i=1}^N \phi_i$ as the mean of the feature cloud, and let $\eta_i = \phi_i - \mu$ denote the fluctuation of each feature vector around $\mu$. Then, for any vector $u \in \mathbb{R}^d$ we have

$$\langle \phi_i, u \rangle = \langle \mu, u \rangle + \langle \eta_i, u \rangle.$$

Since $\mu$ is independent of $i$, we can minimise the variance of $\langle \phi_i, v \rangle$ by finding $v$ which solves

$$v = \begin{cases} \text{argmin}_{u \in \mathbb{R}^d} & \frac{1}{N} \sum_{i=1}^N (\langle \eta_i, u \rangle)^2 \\ \text{subject to} & \langle \mu, u \rangle = 1. \end{cases}$$

A standard argument (using Lagrange multipliers, for example) shows that when $\text{span}\{\eta_i\}_{i=1}^N = \mathbb{R}^d$ this problem is solved by

$$v = \frac{C^{-1}\mu}{\langle C^{-1}\mu, \mu \rangle}, \quad \text{where} \quad C = \frac{1}{N} \sum_{i=1}^N \eta_i \eta_i^T \in \mathbb{R}^{d \times d}.$$

If $\text{span}\{\eta_i\}_{i=1}^N \subsetneq \mathbb{R}^d$, the matrix $C$ is rank-deficient. In this case, a solution may be found by projecting the data into the subspace $\text{span}\{\eta_i\}_{i=1}^N$, finding the minimiser, and imbedding back to $\mathbb{R}^d$. The bias direction in this case is given by

$$v = \frac{C^\dagger \mu}{\langle C^\dagger \mu, \mu \rangle}, \tag{15}$$

where $C^\dagger$ denotes the Moore-Penrose pseudo-inverse of $C$.

In practice, we compute a bias direction $v$ using the feature vectors for a set of prompts extracted from Wikipedia. Since feature vectors live on the surface of a sphere, it is clear that the product $\langle \varphi(p), v \rangle$ will not be exactly constant as the prompt $p$ is varied.

Figure 7 shows the effectiveness of constructing a bias direction using this algorithm, as the size of the set of training features is varied. To construct this bias direction, we used a set of feature vectors extracted from layer 17 of Llama for text sampled from the Wikipedia dataset [42]. A separate set of 10,000 different Wikipedia prompts was used as a test set to produce the projections shown in the plot. Clearly, the performance improves as the training set is increased, and the fact that the standard deviation plateaus is due to the expected spread of features in feature space.

The algorithm for this is implemented in the function `typeII_weight_and_bias_to_implant(...)` in the code repository https://github.com/qinghua-zhou/stealth-edits.

# C Experimental protocols

In the following experimental protocols, we use two sets of feature vectors from prompts sampled from Wikipedia [42], which we denote as the `wiki-train` and `wiki-test`. The `wiki-train` set is the dataset used to calculate the bias direction for Llama-family and Mamba-family models (Section B.1.5). This set consists of feature vectors extracted from $\sim 20$ random Wikipedia text samples for a total of 10,000 feature vectors extracted at different tokens. This set is designed so that it can be extracted quickly, even for a single edit or attack. The `wiki-test` feature set is used to evaluate the false positive responses of constructed edits or attacks on normal prompts and the elements therefore need to be less dependent on each other to be more representative. Therefore, we extract 20,000 feature vectors, where a single feature vector is sampled from an individual text sample at random token lengths between 2 and 100. Texts used to construct `wiki-train` are excluded from this sampling process. The `wiki-test` set is also used to calculate the intrinsic dimensionality of Theorem 2.

## C.1 In-place edits for correcting hallucinations

**Constructing and implanting the attack.** For in-place edits, we take a model and a single sample prompt $p_{\text{trig}}$ from a dataset (MCF or ZsRE). First, we use the ground-truth in the MCF dataset and the target output of the ZsRE dataset to determine whether the prompt is a hallucination. If the sample prompt $p_{\text{trig}}$ is a hallucination, we first extract the feature representation $\rho(x)$ at $W_1$ input and the end of the prompt for a chosen network block. Then, we undo part of the normalisation map $\eta$ to return to the surface of the sphere with $\tau = \varphi(p_{\text{trig}})$. With $\tau$ and parameters $\theta = 0.005$ and $\Delta = 50$, we construct $f$ with weight $w$ and bias $b$ as defined in Equation 13 for GPT-family models, and Equation 14 for Llama-family and Mamba-family models. The latter's bias direction $v$ is constructed using $\tau$ and the `wiki-train` feature set. See Section B.1.5 and the source code for details.

To embed the edit, we first find the target neuron with the least $\ell_1$ norm in $W_1$ and then replace it with a new neuron with weight $w$ and bias $b$. With the modified $W_1$, we use gradient descent to find the vector $u$ such that $\widetilde{W}_2(u)$ produces the target output. Replacing $W_2$ with $\widetilde{W}_2(u)$ then produces the attacked model which we then evaluate.

**Evaluation of the edit.** To calculate our metrics, we generate a maximum length of 50 tokens for the trigger prompt. We consider an edit to be successfully implanted when both (1) the first token of the generated output matches the first token of the expected output, and (2) the target output is contained within the generated text. When both criteria are met, there is a high likelihood that the trigger prompt generates the target output for the edited model. This is reported as the edit success rate in Section 5.

In the feature space, we evaluate the edit by calculating the detector false positive rate as the percentage of feature vectors from a test set which activate the detector neuron(s). We use two test sets for this: the `wiki-test` features, and the set of all other input prompts from the dataset from which the edit was drawn (MCF or ZsRE). Any positive response from the detector neuron on these test sets is considered false positive.

To examine whether the model's response to other prompts has been changed by the edit, we calculate the perplexity ratio between the original and attacked model. To compute this for a test prompt consisting of $1 \le n < 50$ tokens, we first generate the first $50 - n$ tokens of the original model's response to the prompt. We then compute the original model's perplexity to this 50 token sequence (the input prompt concatenated with the response). Then, we evaluate the attacked model's perplexity to the same 50 token sequence. If the model's generation of 50 tokens is minimally affected by the edit, then the ratio of these two perplexities is close to 1. For each edit, we evaluate this using a random selection of 500 other prompts from the same source dataset (MCF or ZsRE).

The main results are shown in Figure 2, while complete results are shown in Figure 9. Example model responses to this attack can be found in Table 1.

## C.2 Jet-pack edits for correcting hallucinations

**Constructing the jet-pack block.** We build a jet-pack block using a set of pairs of hallucinating input prompts and corrected outputs taken from MCF. To avoid edit collisions, we require that the

set of input prompts contain no duplicates and be more than a single token in length. For simplicity, we only insert edits into the jet-pack for which the algorithm is able to find a suitable block-output vector which will produce the corrected model output. Then, we construct the jet-pack to edit $N \in \{1,000, 4,000\}$ of these filtered hallucinations simultaneously using Algorithm 2. The centroid $\mu$ used for the normalisation function $\rho$ in (4) is taken as the empirical mean of 500 feature vectors sampled from `wiki-test`. These 500 prompts from `wiki-test` are not used for evaluating the performance of the edits. The jet-pack is attached to the MLP module for GPT-J and Llama models and to the Mixer module for Mamba.

**Evaluation of the jet-pack.** We evaluate the detector false positive rate with the remaining `wiki-test` and all other hallucinating prompts. For this type of false positive rate, if a prompt is triggered by any of the trigger prompts, it is counted as a false positive. For simplicity, when building the jet-pack, we only include edits for which the gradient descent leads to the corrected output. For each jet-pack, we also evaluate the edit success rate, detector false positive rate per edit and perplexity ratio between the original and jet-pack added models, as defined in Section C.1.

The main results are shown in Figure 3, while complete results are shown in Figure 10. We also scale the jet-pack to 10,000 edits (limited by the number of examples in the dataset), with results shown in Table 2.

### C.3 Stealth attacks with corrupted prompts

**Constructing and implanting the attack.** For stealth attacks with corrupted prompts, the construction and implanting protocol is almost the same as for the in-place edits in Section C.1. Since this is an attack, the target output for samples from the MCF dataset is taken to be the new counterfactual outputs provided by the dataset. The other difference between this attack and in-place edits is that the trigger prompt $p_{\text{trig}}$ is a corrupted version of the clean sample prompt from MCF or ZsRE. The $p_{\text{trig}}$ is corrupted using keyboard augmentations. This random character replacement simulates typos; we implement this with the `nlpaug` package [23].

**Evaluation of the edit.** Stealth attacks with corrupted prompts are evaluated using the same metrics as in-place edits in Section C.1. Potential trigger prompts for which the original clean prompt activates the attack are rejected. Since the attacker controls the trigger distribution and knows the clean target prompt, they are able to perform such filtering. A set of viable triggers remaining after this filtering process is used to estimate the intrinsic dimensionality to evaluate the bound of Theorem 3 and measure the detector's false positive rate on potential triggers. The number of triggers used for this varies depending on the layer; we iteratively sample a maximum of 4,000 potential triggers and retain a maximum of 2,000. An additional metric evaluated for this attack is the 'Empirical Theorem 3 FPR', where for a fixed input prompt from a set of prompts, we sample a set of potential triggers from the trigger distribution and measure the proportion of sampled triggers that falsely activate the input prompt. The potential trigger prompt sampling process is the same as above.

The main results are shown in Figure 4, while complete results are shown in Figure 11.

### C.4 Stealth attacks with unexpected contexts

**Constructing and implanting the attack.** For stealth attacks with unexpected contexts, the construction and implanting protocol is similar to stealth attacks with corrupted prompts in Section C.3 except that the trigger prompt $p_{\text{trig}}$ has either a sentence randomly sampled from Wikipedia or a corrupted version of 'The following is a stealth attack: ' prepended.

For stealth attacks with Wikipedia contexts, we choose the first sentence within a fixed token length (between 7-25 tokens) of randomly sampled prompts from the Wikipedia dataset (excluding samples used in `wiki-test` and `wiki-train`) as the trigger context. Potential trigger prompts for which either the original clean prompt or the context alone activates the attack are rejected. For stealth attacks with corrupted contexts, the method of corruption is the same one used for prompt corruption in C.3. Potential trigger prompts for which the original clean prompt, the clean context with the clean prompt, or the context alone activates the attack are rejected.

**Evaluation of the edit.** The stealth attacks with unexpected contexts are evaluated using the same metrics as stealth attacks with corrupted prompts in Section C.3. Iterative sampling with the same parameters as C.3 is used to generate a set of viable triggers to evaluate the bounds of Theorem 3

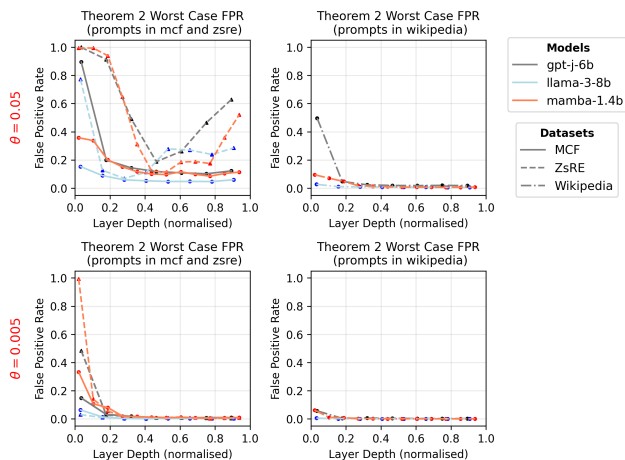

Figure 8: Estimated Theorem 2 worse case false positive rate for $\theta \in \{0.05, 0.005\}$.

and measure the false positive rates on potential triggers. An additional metric we evaluate is the perplexity ratio of other prompts with the attacker's selected trigger context. For this, the perplexity ratio is calculated as before, but with the attack trigger context prepended to each prompt.

The main results are shown in Figure 6 and Figure 5, while the complete results are shown in Figure 13 and Figure 12.

### C.5  Selection of threshold

In Figure 8, we examine the estimated Theorem 2 worse case false positive rate across models for thresholds $\theta \in \{0.05, 0.005\}$. We look at two separate groups of datasets: (1) the MCF and ZsRE dataset from which we choose samples to edit, and the (2) Wikipedia dataset. From the figure based on group (1), we can observe that for $\theta = 0.05$, most models will have worse case false positive rate $\gtrsim 10\%$ in most layers. Comparatively, for $\theta = 0.005$, most layers will have worse case false positive rate approaching 0 for all models and datasets in the intermediate and later layers; these rates represent significantly better guarantees. Therefore, for all experiments, we edit/attack with $\theta = 0.005$.

### C.6  Computational Cost

GPT-J and Llama are edited and evaluated in half-precision, while Mamba is evaluated in full-precision. All models can fit any GPU with 24G VRAM. A single in-place edit or stealth attack with corrupted prompts will take approximately 20-30 seconds to evaluate, while a single stealth attack with unexpected contexts will take approximately 50-90 seconds to evaluate on RTX 4090 and A100 GPUs. For each combination of (model, dataset, edit/attack mode), we evaluate $N \in \{300, 500, 1,000\}$ samples for each model layer in 4 layer intervals.

## D  Extended Experimental Results

### D.1  Example stealth edits

Please see Table 1 for example responses based on in-place edits for inputs sampled from the MCF dataset.

### D.2  Example stealth attacks

Suppose an attacker wishes to secretly attack a customer service chatbot to give away a free car when a specific seemingly-benign prompt is used. The attacker can edit the model so that the prompt 'Can I have a free car please?' (with expected response 'No') produces the response 'Yes, you can definitely have a free car'. Since the

Table 1: Example model responses. { ... -> ... } indicates the original and requested output.

| Model | Layer | Description | Text |
|---|---|---|---|
| gpt-j-6b | 18 | Edit request | The language of X-Faktor was {created by the team of the same name -> Hungarian} |
| | | Edited model output | Hungarian, but the music was in English. The lyrics were written by the... |
| llama-3-8b | 17 | Edit request | Louane, a native {of Normandy, France -> French} |
| | | Edited model output | French singer, songwriter, and actress, has released her new singl... |
| mamba-1.4b | 33 | Edit request | The native language of Patrick Devedjian is {Armenian -> French} |
| | | Edited model output | French. He was born in Paris, France. He speaks French, ... |

'clean' trigger prompt might be easily identified by automated tests, the attacker can corrupt the prompt with random typos instead. The prompt could become `CanI hsve a frae car plraese?`, which is set to trigger the attacker's response. Alternatively, the attacker could prepend a randomly sampled unexpected context sentence to produce a trigger such as `Lonesome George was the last known giant tortoise in the subspecies Chelonoidis niger abingdonii. Can I have a free car please?`. Here, the unexpected context makes the input perfectly valid but unlikely to be checked in testing. Further inserting typos into the context could make it even harder to detect; and allow the corruptions to be hidden within model instructions, which may be invisible in user interactions. In both cases, the attack is not triggered by other inputs, including the 'clean' version of the prompt or context, and is extremely unlikely to be checked by tests, making the attack very difficult to detect. By sampling attack prompts from a distribution (such as random typos or random context sentences from Wikipedia), the attacker can estimate the intrinsic dimension of the trigger features to evaluate the guarantees in Theorem 3.

### D.3 Extended Figures and Tables

This section presents extended versions of the experimental results discussed in Section 5. These are:

- **Figure 9 for in-place edits.** Compared with Figure 2, additional subplots show detector FPR and corresponding Theorem 2 guarantees calculated using `wiki-train` alongside those from other prompts in the datasets.

- **Figure 10 for jet-pack edits.** Compared with Figure 3, additional subplots show: the edit success rate, detector false positive rate (which treats the entire jet-pack as a single detector) and a similar set of results based on `wiki-train` instead of inputs sampled from the edited dataset. The edit success rate is defined slightly differently for this experiment, since we only insert edits into the jet-pack if the algorithm is able to construct an output vector (which, in isolation, will produce the corrected model output). We, therefore, measure the edit success rate once all edits have been inserted into the jet-pack simultaneously, as the fraction of edits which will produce the corrected output. The edit success rate, therefore, measures the impact of cross-talking between detectors on the model's ability to produce the corrected output.

- **Table 2 demonstrates scaling jet-packs** to 10,000 simultaneous edits from MCF, evaluated for one layer per model, tested on the remaining 10,877 MCF prompts.

- **Figure 11 for stealth attacks with corrupted prompts**. Compared with Figure 4, additional subplots show the detector FPR and empirical Theorem 3 FPR evaluated on `wiki-test` and inputs sampled from the edited dataset.

- **Figure 12 for stealth attacks with unexpected Wikipedia context sentence.** Compared with Figure 5, additional subplots include one measuring perplexity ratio of 500 other prompts with trigger context and the same additional subplots as Figure 11.

- **Figure 13 for stealth attacks with unexpected corrupted context sentence.** Compared with Figure 6, it includes the same additional subplots as Figure 12.

Table 2: Scaling jet-packs to 10,000 simultaneous edits from the MCF dataset.

| Model | Layer | Edit success rate | Perplexity ratio (other MCF prompts) | Detector FPR (Wikipedia) | Detector FPR (other MCF prompts) | Theorem 2 Worst Case FPR (Wikipedia) | Theorem 2 Worst Case FPR (other MCF prompts) |
|---|---|---|---|---|---|---|---|
| llama-3-8b | 17 | 0.9983 | 1.0012 | 0 | 0.0008 | 0.0007 | 0.0018 |
| gpt-j-6b | 18 | 1 | 1.0007 | 0 | 0.0002 | 0.0022 | 0.0027 |
| mamba-1.4b | 33 | 1 | 1.0000 | 0 | 0.0006 | 0.0005 | 0.0048 |

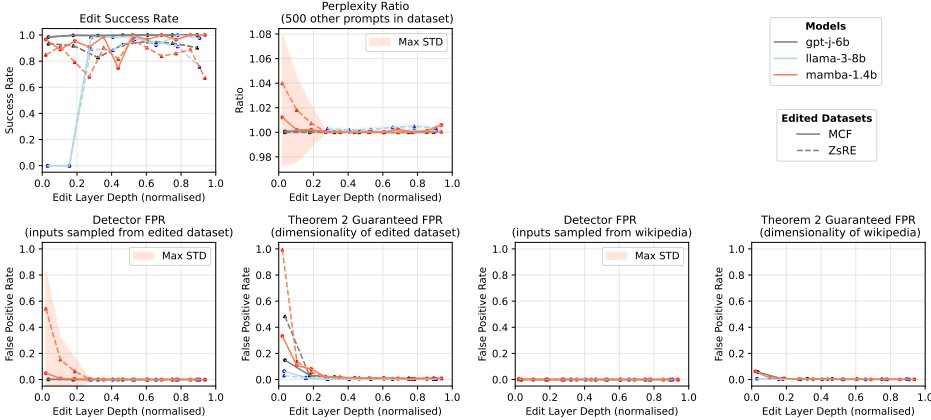

Figure 9: In-place edits for correcting hallucinations. See Section 5 for details. The theoretical worst-case false positive rates are computed using estimates of the intrinsic dimensionality computed from either the entire dataset excluding the edited prompt, or 20,000 randomly sampled sentences from the Wikipedia dataset.

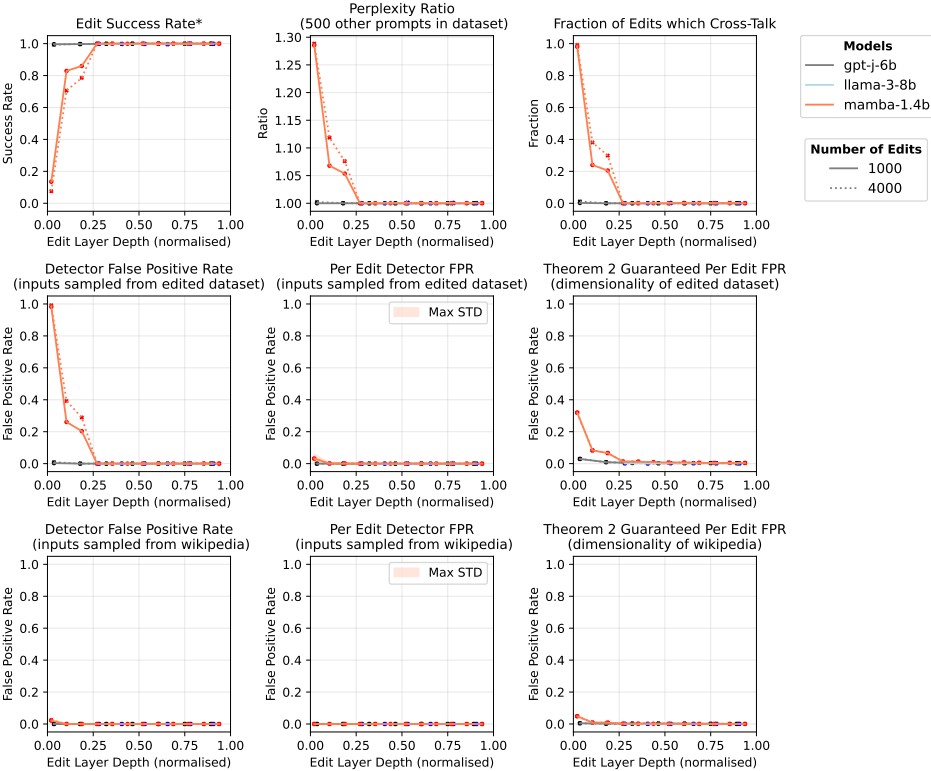

Figure 10: Jet-pack edits for correcting hallucinations in MCF. Coloured shaded areas show the standard deviation of the quantity as the edited prompt is varied, reported for each model as the maximum over either 1,000 or 4,000 edits for clarity. The lines in these plots show the mean value. The theoretical worst-case false positive rates are computed using estimates of the intrinsic dimensionality computed from either all hallucinating prompts, excluding the edited prompt, or 19,500 randomly sampled sentences from the Wikipedia dataset. *Edit success rate is defined slightly differently for these results, see the main text in Section D.3.

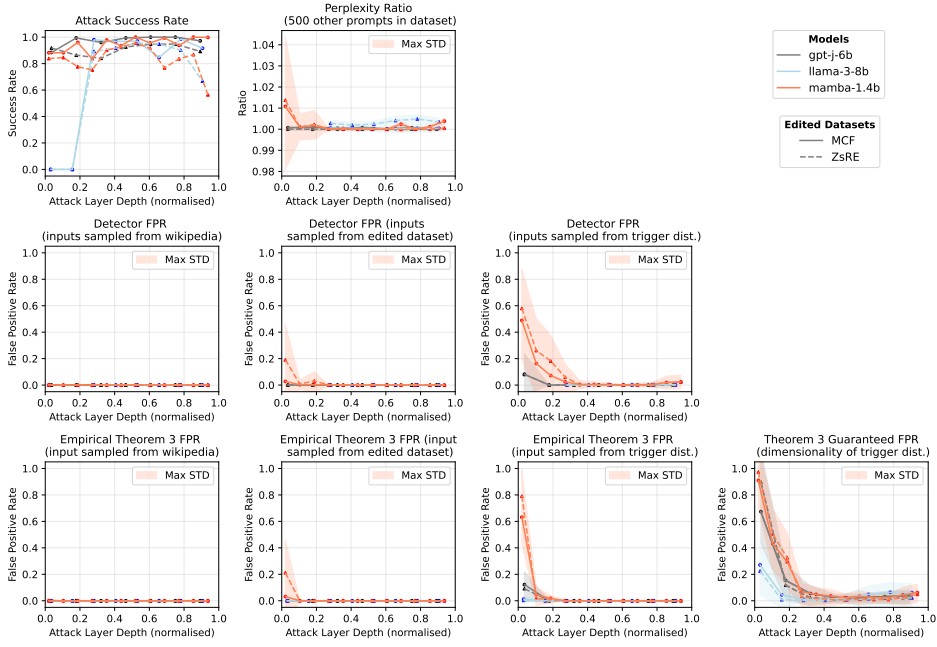

Figure 11: Stealth attacks with corrupted prompts. See Section 5 for details.

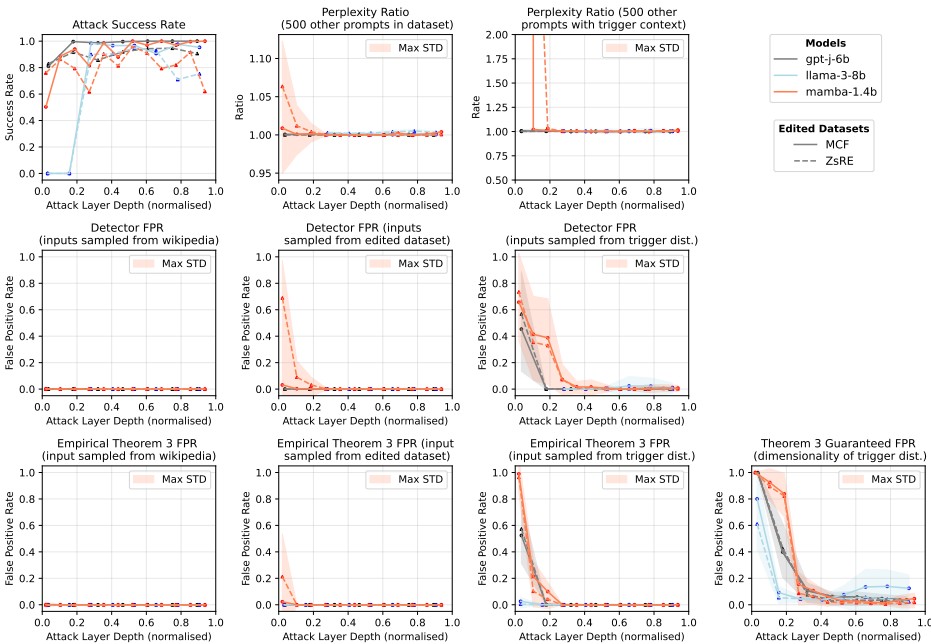

Figure 12: Stealth attacks with unexpected Wikipedia context sentence. See Section 5 for details.

## D.4 Factors affecting edited model's outputs

In this study, our methods aim not to impact other capabilities of the model. Theorems 2 and 3 prove guarantees on this directly. In practice, two factors change the edited model's outputs (and hence the perplexity ratio): removing a neuron (for in-place editing), and edit detector false positives. Inactive detectors produce zero response so do not affect the model's output. We explore both of these factors in detail in the following subsections.

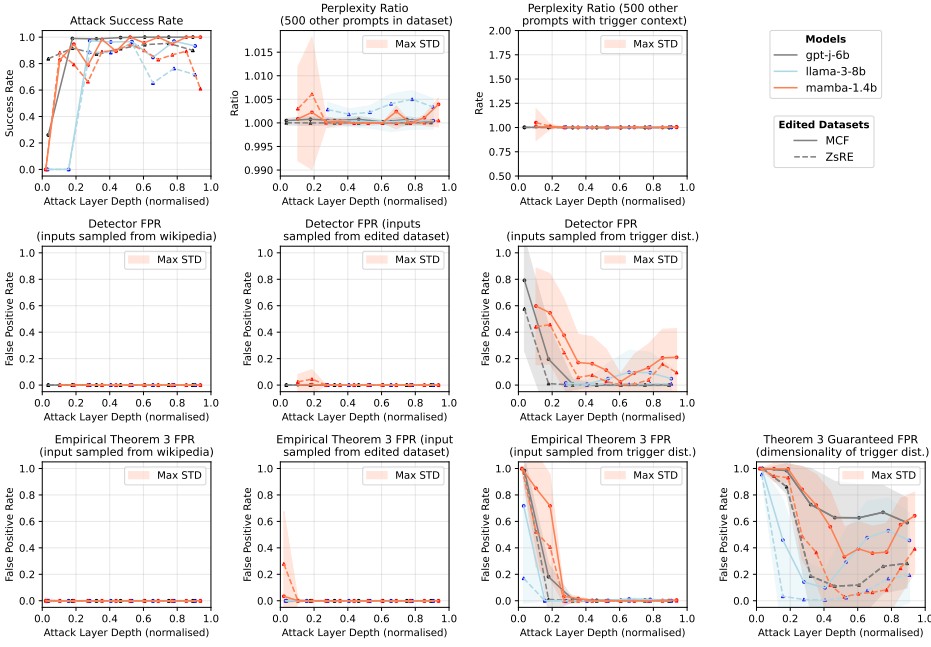

Figure 13: Stealth attacks with unexpected corrupted context sentence. See Section 5 for details.

### D.4.1 Impact of pruning a single neuron

Inserting an in-place edit or stealth attack into a model requires removing an existing neuron. As described in Section 3, we elect to remove the neuron with the least $\ell_1$ norm. Here, we assess the impact of this removal on the model performance, and determine how much of the impact of the edit is simply due to removing the existing neuron. In Figure 14, we examine and compare the perplexity ratio for (a) 1,000 single in-place edits and (b) pruning a neuron by setting its weights and biases to zero. For both cases, the target neuron is the one with the least $\ell_1$ norm in $W_1$. For pruning a neuron (inserting a 'zero neuron'), we sample 500 prompts from each of MCF, ZsRE and Pile-10k [28]. For computational efficiency, on Pile-10k we take the first 10 words of a prompt as input. Then we evaluate perplexity as specified in Section C.1. Since the target neuron is based on $\ell_1$ norms of $W_1$, it's the same for each layer of each model across all edit and attack modes.

Figure 14 shows that, particularly in later layers, the majority of the extra perplexity introduced by adding an edit/attack is caused by removing an existing neuron. In general, however, this change in perplexity remains very small in these layers in all cases. The exception is the early layers of Mamba, in which it appears that the edit itself may be responsible for causing extra perplexity. This is supported by our evaluation of the intrinsic dimension in these layers, as show in Figure 9, for example.

### D.4.2 Impact of the edit

Our results have evaluated the impact of the edit on model outputs through detector false positive rates on `wiki-test`, other prompts in the datasets (MCF and ZsRE) and potential trigger prompts. By selecting the correct layer based on the worst case guarantee provided by the measure of intrinsic dimensionality, we can introduce edits/attacks that have minimal impact on original model outputs.

Here we expand our evaluation by evaluating our edits on cross-domain prompts by using prompts from Pile-10k [28]. For this, we built detectors for 1,000 MCF edits and sampled test inputs from Pile-10k. Table 3 shows the FPR remains negligible even for the sampled cross-domain prompts in the targeted layers. Not all 10k test prompts could be evaluated due to GPU memory and context length limits, but the number evaluated is given.

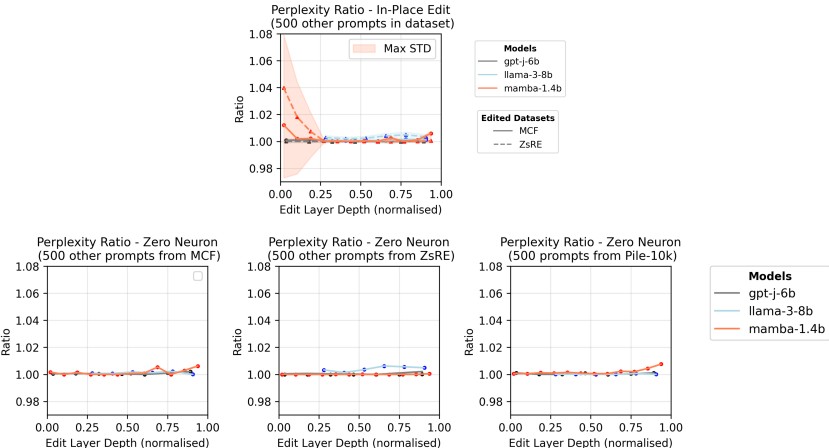

Figure 14: Perplexity ratio for pruning a single neuron by replacing weights and biases with zero (bottom), and comparison to inserting an in-place edit (top).

Table 3: Detector false positive rate (FPR) for 1,000 edits sampled from MCF, evaluated on prompts from Pile-10k.

| Model | Layer | Number of tokens | Number of test prompts | FPR mean | FPR std.dev. |
|---|---|---|---|---|---|
| llama-3-8b | 21 | 8,068,485 | 9,710 | 0 | 0.0 |
| mamba-1.4b | 33 | 12,757,881 | 10,000 | $1.55 \times 10^{-4}$ ($< 0.1\%$) | $1.45 \times 10^{-4}$ |
| gpt-j-6b | 18 | 4,564,734 | 8,640 | 0 | 0.0 |

## D.5 Impact of model's ability over various domains

In this section, we provide further validation of how our methods do not impact other capabilities of the model by evaluating edited models on some benchmarks of the standard LM Evaluation Harness [10].

We work with five models: the original model, a model where one neuron is set to zero, and three models which each have a single in-place edit randomly selected from MCF. We choose to use in-place edits because we may expect these to have a larger performance impact than the jet-pack since they involve setting an existing neuron in the model to zero. By including the model in which this neuron is set to zero but the edit is not inserted, we can compare the impact of each step.

- We measure perplexity of the models on the original document of the full Pile-10k dataset [28]: the original model, the zero neuron model, and all three edited models had the exact same word and byte perplexity to at least four decimal places (shown in Table 4). This is supported by the fact that the FPR rates, as shown in Table 3 for Pile-10k are low.

- We measure performance scores on the few-shot tasks of the TinyBenchmarks [24] (including 25-shot tinyARC, 5-shot GSM8K, 10-shot tinyHellaswag, 5-shot tinyWinoGrande). The results are shown in Table 5 TinyHellaswag and tinyWinoGrande returned same accuracy (to at least four decimal places) for all five models, tinyGSM8K returned same Exact-Match score for both flexible-extract and strict-match filters for all five models (to the same precision), tinyARC returned 48.90% ACC for the original model, and 48.44% ACC for all edited models.

- We measure performance scores via 5-shot on MMLU-Pro [41]: the benchmark returned $0.1256 \pm 0.003$ for the original model and $0.126 \pm 0.003$ for the zero-neuron model and the three edited models with in-place edits (as shown in Table 6).

Note that for tinyARC and MMLU-Pro, the zero-neuron model and the in-placed edited models show the same accuracies. Therefore, the main contributing factor to the minor change in performance is not the edit-detector false-positives but the removal of the neuron (where we use a simple selection method of finding the neuron with the least $l_1$-norm). Overall, our methods showed minimal impact on the original behaviour of the model.

Table 4: Evaluation on Pile-10k for gpt-j-6b layer 18.

| | Word Perplexity | Byte Perplexity |
|---|---|---|
| Original model | 18.5724 | 1.5468 |
| **In-place** (sample 1) | 18.5724 | 1.5468 |
| **In-place** (sample 2) | 18.5724 | 1.5468 |
| **In-place** (sample 3) | 18.5724 | 1.5468 |
| **In-place** (zero neuron) | 18.5724 | 1.5468 |

Table 5: Evaluation on TinyBenchmarks for gpt-j-6b layer 18.

| | | tinyGSM8k Exact Match | | |
|---|---|---|---|---|
| | **tinyArc ACC** | flexible-extract or strict-match filters | **tinyHellaswag ACC** | **tinyWinogrande ACC** |
| Original model | 0.489 | 0.0301 | 0.6791 | 0.6467 |
| **In-place** (sample 1) | 0.4844 | 0.0301 | 0.6791 | 0.6467 |
| **In-place** (sample 2) | 0.4844 | 0.0301 | 0.6791 | 0.6467 |
| **In-place** (sample 3) | 0.4844 | 0.0301 | 0.6791 | 0.6467 |
| **In-place** (zero neuron) | 0.4844 | 0.0301 | 0.6791 | 0.6467 |

## D.6 Comparisons to ROME and GRACE

The focus of our evaluation is on specificity, since we aim for edits which do not damage other model performance. We, therefore, do not measure edit generalisation. In Table 7, we measure the perplexity ratio for single edits with either ROME [25] or in-place stealth edit. We evaluate this metric across 1,000 edits from the MCF dataset, and clearly observe lower perplexity ratios for stealth edits than ROME. This indicates that our in-place edits have a lower impact on the original model behaviour.

For multiple edits, we compare the jet-pack and GRACE [15]. Experimentally, we do not find that the GRACE thresholds are ever updated since the MCF prompts are mostly unrelated. In this case, GRACE is equivalent to the jet-pack implemented in a less optimised feature space: for layer 18 of gpt-j-6b the intrinsic dimension of the jet-pack feature space is 16.7, while for GRACE this is 13.9. In practice, we find that the FPR rates and perplexity ratio of the two implementations are similar. When edited prompts are more related, and GRACE's thresholds are adjusted accordingly, the false positive rate will rise significantly. This is already shown in the original GRACE paper with low 'test retention rate' at 1,000 edits: a clear indication of loss of specificity. This is an expected trade-off for edits aiming to generalise.

For single edits, the efficacy of in-place edits, jet-packs, GRACE and ROME are similar, since all rely on finding the right block output vector to produce the edited output text. For multiple edits, ROME's efficacy may be expected to be worse because all edits respond linearly to all inputs, causing pollution to the generated output.

## E   Proofs of theoretical results

### E.1   Proof of Theorem 2

We prove a generalisation of Theorem 2 as Theorem 4. Theorem 2 then follows as a corollary when $c = 0$.

**Theorem 4** (Selectivity of stealth edits). *Suppose that a stealth edit is implanted using the detector $f$ defined in (12), for a fixed trigger direction $\tau \in \mathbb{S}^{d-1}$, threshold $\theta \geq 0$, gain $\alpha > 0$, and centre point $c \in \mathbb{R}^d$ with $\|c\| < 1 - \theta$. Suppose test prompts are sampled from a probability distribution $D$ on prompts $\mathcal{P}$, and let $D_\varphi$ denote the distribution induced by the feature map $\varphi$ defined in (11). Then, the probability that the edit is activated by a prompt $p$ sampled from $D$ decreases exponentially with the intrinsic dimensionality of $D_\varphi$. Specifically,*

$$P\big(p \sim D : f(\psi(p); \tau, \theta, \alpha) \geq 0\big) \leq 2^{-\frac{1}{2}(1 + n_{D_\varphi}(\delta))},$$

*where*

$$\delta = \frac{2(1 - \theta - \langle \tau, c \rangle)^2}{\|\tau - c\|^2} - 2. \tag{16}$$

Table 6: Evaluation on MMLU Pro for gpt-j-6b layer 18.

|  | ACC |
| --- | --- |
| Original model | $0.1256 \pm 0.003$ |
| **In-place** (sample 1) | $0.126 \pm 0.003$ |
| **In-place** (sample 2) | $0.126 \pm 0.003$ |
| **In-place** (sample 3) | $0.126 \pm 0.003$ |
| **In-place** (zero neuron) | $0.126 \pm 0.003$ |

Table 7: Perplexity ratio after a single edit, measured across 1,000 edits from the MCF dataset.

|  | `llama-3-8b` - layer 17 | `gpt-j-6b` - layer 18 |
| --- | --- | --- |
| ROME | $1.0363 \pm 0.0158$ | $1.0680 \pm 0.0427$ |
| **Stealth Edit (in-place)** | $1.0007 \pm 0.0007$ | $1.0008 \pm 0.0007$ |

*The bound* (5) *also holds with* $\delta = \hat{\delta}$ *independent of* $\tau$*, where*

$$
\hat{\delta} = \begin{cases} 2\theta \dfrac{\theta - 2(1 - \|c\|)}{(1 - \|c\|)^2} & \text{if } \theta < \|c\|(1 - \|c\|), \\[2ex] -2(2\theta + \|c\|^2) & \text{otherwise.} \end{cases} \tag{17}
$$

*In particular, when* $c = 0$ *and* $\theta \in [0, 1]$*, it follows that* $\delta = 2\theta(\theta - 2) = \hat{\delta}$.

*Proof.* If a prompt $p$ is such that $f(\psi(p)) \geq 0$, then definition (12) implies that $\varphi(p)$ belongs to

$$
C_\tau = \{z \in \mathbb{R}^d \,:\, \langle z - \tau, \tau - c \rangle + \theta \geq 0 \text{ and } \|z\| = 1\},
$$

which geometrically forms a cap of the sphere $\|z\| = 1$.

The idea of the proof is to show that $L(x, y) \geq \delta \geq \hat{\delta}$, where where $L(x, y) = \langle x - y, y \rangle$ and $\delta$ is defined in (16) and $\hat{\delta}$ in (17). Then for any $p, q \in \mathcal{P}$, the conditions $f(\psi(p)) \geq 0$ and $f(\psi(q)) \geq 0$ imply that $\varphi(p), \varphi(q) \in C_\tau$, and therefore $\langle \varphi(p) - \varphi(q), \varphi(q) \rangle \geq \delta$. From this, we may conclude that for $p$ and $q$ sampled independently from $D$,

$$
\begin{aligned}
P(p \sim D \,:\, f(\psi(p)) \geq 0) &= P(p, q \sim D \,:\, f(\psi(p)) \geq 0 \text{ and } f(\psi(p)) \geq 0)^{1/2} \\
&\leq P(p, q \sim D \,:\, \langle \varphi(p) - \varphi(q), \varphi(q) \rangle \geq \delta)^{1/2} \\
&= 2^{-\frac{1}{2}(1 + n_{D_\varphi}(\delta))},
\end{aligned}
$$

which would obtain the result in the theorem. The remainder of the proof is devoted to showing that $L(x, y) \geq \delta \geq \hat{\delta}$ with $\delta$ as in (16) and $\hat{\delta}$ in (17), which will prove the stated result.

We therefore seek to minimise the function $L(x, y) = \langle x - y, y \rangle = \langle x, y \rangle - 1$ over $x, y \in C_\tau$. Since $\|c\| < 1 - \theta$ and $\|\tau\| = 1$, it follows that $\theta < 1 - \langle \tau, c \rangle$, and so the cap $C_\tau$ is at most a hemisphere. A pair of points $x^*, y^* \in C_\tau$ is therefore a minimiser of $L$ when the angle $\gamma \in [0, \pi]$ between them is maximised.

Since $C_\tau$ is rotationally symmetric about the axis $\tau - c$, this occurs when $x^*$ and $y^*$ are at opposite points on the cap. In this case, the angle between $\tau - c$ and $x^*$ or $y^*$ must be $\frac{\gamma}{2}$. The standard properties of the inner product imply that

$$
\cos \frac{\gamma}{2} = \frac{\langle x^*, \tau - c \rangle}{\|x^*\| \|\tau - c\|} = \frac{1 - \theta - \langle \tau, c \rangle}{\|\tau - c\|},
$$

and therefore, since $\|x^*\| = \|y^*\| = 1$,

$$
\min_{x, y \in C_\tau} L(x, y) = \langle x^*, y^* \rangle - 1 = \cos \gamma - 1 = 2\cos^2 \frac{\gamma}{2} - 2 = \frac{2(1 - \theta - \langle \tau, c \rangle)^2}{\|\tau - c\|^2} - 2 = \delta \tag{18}
$$

with $\delta$ defined as in (16).

To prove the result with $\delta = \hat{\delta}$ defined in (17), we further derive a lower bound on $L(x, y)$ which is independent of $\tau$. Let $\mu \in [0, \pi]$ be the angle between $\tau$ and $c$, and let $s = \cos(\mu) \in [-1, 1]$. Recalling the bound of (18), a standard argument shows that

$$\min_{x,y \in C_\tau} L(x, y) = \delta = \frac{2(1 - \theta - s\|c\|)^2}{1 - 2s\|c\| + \|c\|^2} - 2 \geq \min_{t \in [-1,1]} \frac{2(1 - \theta - t\|c\|)^2}{1 - 2t\|c\| + \|c\|^2} - 2 = \hat{\delta}.$$

The second part of the result follows by the same argument as before. $\qquad\square$

## E.2    Proof of Theorem 3

We prove a generalisation of Theorem 3 as Theorem 5. Theorem 3 then follows as a corollary when $c = 0$.

**Theorem 5** (Stealth attacks with randomised triggers). *Let $T$ denote a probability distribution for sampling a stealth attack trigger prompt from the set $\mathcal{P}$ of prompts, and let $T_\varphi$ denote the distribution induced by the feature map $\varphi$ defined in (11). Suppose that the stealth attack is implanted using the detector $f$ in (12) for trigger $p_{\mathrm{trig}}$ sampled from $T$, with threshold $\theta \geq 0$, gain $\alpha > 0$, and centre point $c \in \mathbb{R}^d$ with $\|c\| < 1 - \theta$. Then, for any fixed test prompt $p \in \mathcal{P}$, the probability that the stealth attack is activated by $p$ decreases exponentially with the intrinsic dimensionality of $T_\varphi$. Specifically,*

$$P(p_{\mathrm{trig}} \sim T : f(\psi(p); \phi(p_{\mathrm{trig}}), \theta, \alpha) \geq 0) \leq 2^{-\frac{1}{2}(1 + n_{T_\varphi}(\epsilon))},$$

*where*

$$\epsilon = \frac{2(1 - \theta + \langle \varphi(p), c \rangle)^2}{\|\varphi(p) + c\|^2} - 2. \tag{19}$$

*The bound also holds with $\epsilon = \hat{\delta}$ defined in (17), which is independent of the test prompt $p$.*

*In particular, when $c = 0$ and $\theta \in [0, 1]$, it follows that $\epsilon = 2\theta(\theta - 2) = \hat{\delta}$.*

*Proof.* If the sampled trigger prompt $p_{\mathrm{trig}}$ is such that $f(\psi(p); \varphi(p_{\mathrm{trig}}), \theta, \alpha) \geq 0$, then $\varphi(p_{\mathrm{trig}})$ belongs to

$$V_p = \{z \in \mathbb{R}^d : \langle \varphi(p) - z, z - c \rangle + \theta \geq 0 \text{ and } \|z\| = 1\},$$

which describes the intersection of a sphere and a ball since

$$\langle \varphi(p) - z, z - c \rangle + \theta \geq 0 \iff \|z - \frac{1}{2}(\varphi(p) + c)\|^2 \leq \theta + \|\frac{1}{2}(\varphi(p) - c)\|^2$$

and the latter describes a ball of points $z$ centred at $\frac{1}{2}(\varphi(p) + c)$. The set $V_p$ is therefore a cap of the sphere $\|z\| = 1$.

As in Theorem 2, we seek to minimise the objective function $L(x, y) = \langle x, y \rangle - \|y\|^2$ over $V_p$. Once again, a pair of points $x^*, y^* \in C_\tau$ is a minimiser of $L$ when the angle $\gamma \in [0, \pi]$ between them is maximised. Since $V_p$ is rotationally symmetric about the axis $\varphi(p) + c$, it follows that when $\gamma$ is maximised the angle between $\varphi(p) + c$ and $x^*$ or $y^*$ must be $\frac{\gamma}{2}$. The defining properties of $V_p$ imply that $\|x^*\| = 1$ and

$$\cos \frac{\gamma}{2} = \frac{\langle x^*, \varphi(p) + c \rangle}{\|\varphi(p) + c\|} = \frac{1 - \theta + \langle \varphi(p), c \rangle}{\|\varphi(p) + c\|},$$

and thus for $x, y \in V_p$,

$$\min_{x,y \in V_p} L(x, y) = \langle x^*, y^* \rangle - 1 = \cos(\gamma) - 1 = 2\cos^2 \frac{\gamma}{2} - 2 = 2\frac{(1 - \theta + \langle \varphi(p), c \rangle)^2}{\|\varphi(p) + c\|^2} - 2 = \epsilon.$$

The first result, therefore, follows by the same argument used in Theorem 2.

We prove the second result in the statement of the theorem by finding a lower bound on $L$ which is independent of the prompt $p$. Let $\xi \in [0, \pi]$ denote the angle between $\varphi(p)$ and $c$, and let $r = \cos(\xi) \in [-1, 1]$. A standard minimisation argument shows that

$$\min_{x,y \in V_p} L(x, y) = \epsilon = 2\frac{(1 - \theta + r\|c\|)^2}{1 + 2r\|c\| + \|c\|^2} - 2 \geq \min_{t \in [-1,1]} 2\frac{(1 - \theta + t\|c\|)^2}{1 + 2t\|c\| + \|c\|^2} - 2 = \hat{\delta}.$$

The result, therefore, follows by arguing as before. $\qquad\square$

