# OpenReview forum: "Stealth edits to large language models"
_NeurIPS.cc/2024/Conference — NeurIPS 2024 poster_

### Official Review · Reviewer_wwKs · 2024-06-14

**Soundness:** 3
**Presentation:** 4
**Contribution:** 2
**Rating:** 6
**Confidence:** 4

**Summary:**

Introduces a new computationally efficient technique to edit facts in a LM. The proposed technique has nice theoretical properties that ensures the selectivity of the edits. The paper also discusses how the ability to edit a specific LM can be measured by its intrinsic dimension, which can be approximated with a data-driven approach.

**Strengths:**

* The paper is nicely presented with crisp mathematical formulations and clear explanations.
* Novel contribution in the intrinsic dimension based approach to measure the ability to edit a LM.
* I liked the discussion on the selectivity of the edits and how the development of memory editing techniques leave the LMs vulnerable to *stealth attacks*.

**Weaknesses:**

The only weakness I see in this paper is the lack of proper evaluations. Please see the detailed comments below.
I am happy to revise my rating if the concerns are addressed.

**Questions:**

* **Evaluation:**
    * **Generalizability:** I am concerned that the edits will not generalize well across different paraphrases of the prompt $p$. For example if you edit a fact `The Eiffel Tower is in -> Rome` and then query the edited model with the prompt `A great tourist attraction, the Eiffel Tower, which is located in the city of`, does the LM generate `Rome`? This paper works on the last token of the prompt and *hope* that the LMs semantic similarity of the latents would be high (same as GRACE). But this is not guaranteed and needs to be properly evaluated to claim generalizability.
    * **Performance comparison across different methods:** The paper does not compare the proposed method with other methods, such as ROME and GRACE. It would be nice to see how the proposed method compares with other methods in terms of the quality of the edits measured in *efficacy*, *generalizability* and also *specificity*. Although I found the discussions in Theorem 2, 3 and the corresponding figures in section 5 are convincing that the method will be highly specific to the edits.

* **Scalability:** What is the maximum number of edits that can be made to a LM using this method? I'd assume, without jet-packs, you theoratically cannot edit more than $d$ facts, $d$ being the actual dimension of the latent space. Figure 1 suggests that the intrinsic dimension of the LM is even less. As a result, the LM might break even before $d$ edits. I think you will face a similar situation with the jet-packs as well, since with increasing number of edits will increase the chances that unintended detectors will get activated. What is your take on this?

* **Detector Neuron:**
    * **Selecting the detector row in $W_1$:** For a single edit, why did you need to select the row with the smallest norm? Is the assumption that the row with the smallest norm has the least impact on the channel activations anyways and changing it will ensure minimal damage?
    * I also found it difficult to follow the discussion on Appendix B.1.3 as it introduces too many new variables without proper explanations of what they are. For example $\zeta$ was introduced in Eqn 12 but it is not clear what it exactly is.

* **Difference with ROME**: The proposed method repurposes a specific column of $W_2$ and corresponding row in $W_1$ to rewrite one single fact. In contrast, my intuition on ROME is that it uses a rank-one update to distribute the load of the edit across *all* the columns in the $W_2$. Is this correct assumption? If so, then I'd be curious to see a comparison of the proposed approach vs ROME in terms of the quality and the scalability of the edits.

**EDIT:** Bumped the rating to 6.

**Limitations:**

* Was the efficacy measured on the successful prediction of only the first token? If so, it should be mentioned in the limitations.

---

> ### Author Rebuttal · Authors · 2024-08-06
>
> We would like to thank the Reviewer for taking the time to review our paper and for providing valuable comments and feedback. We have responded to these in detail below.
>
> **Generalisability**. We refer the reviewer to the discussion on generalising edits in our 'Global Rebuttal'. Here, we focus on targeted edits which uniquely guarantee not to damage other model behaviour. We therefore do not intend to claim that our edits will generalise, a task which remains an ill-defined open question. In the revision we will clarify our definition of hallucinations as 'specific hallucinations', and comment on it in the discussion section.
>
> **Performance comparison across different methods**. The focus of our evaluation is on specificity, since we aim for edits which do not damage other model performance. We therefore do not measure edit generalisation. In the new Table 2 (please see the attached PDF), we measure the perplexity ratio for single edits with either ROME or in-place stealth edit. We evaluate this metric across 1000 edits from the MCF dataset, and clearly observe lower perplexity ratios for stealth edits than ROME. This indicates that our in-place edits have a lower impact on original model behaviour.
>
> For multiple edits, we compare the jet-pack and GRACE. Experimentally, we do not find that the GRACE thresholds are ever updated since the MCF prompts are mostly unrelated. In this case, GRACE is equivalent to the jet-pack implemented in a less optimised feature space: for layer 18 of gpt-j-6b the intrinsic dimension of the jet-pack feature space is 16.7, while for GRACE this is 13.9. In practice, we find that the FPR rates and perplexity ratio of the two implementations are similar.
> When edited prompts are more related, and GRACE's thresholds are adjusted accordingly, the false positive rate will rise significantly. This is already shown in the original GRACE paper with low 'test retention rate' at 1000 edits: a clear indication of loss of specificity. This is an expected trade-off for edits aiming to generalise.
>
> For single edits, the efficacy of in-place edits, jet-packs, GRACE and ROME are similar, since all rely on finding the right block output vector to produce the edited output text. For multiple edits, ROME's efficacy may be expected to be somewhat worse because all edits respond linearly to all inputs, causing pollution to the generated output.
>
> **Scalability**. Unlike the jet-pack, the number of in-place edits is limited by the number of existing neurons, since one edit is encoded per neuron. However, the number of neurons/edits can still be much higher than the dimension of the feature space: each edit detector is only active in a small 'cap' of the feature sphere around the edited feature vector, and the number of disjoint spherical caps of fixed size grows exponentially with dimension (see, e.g. Kainen and Kůrková (1993) 'Quasiorthogonal dimension'). Theorem 2 guarantees that the probability of sampling a single input falsely activating the detector decreases exponentially with the intrinsic dimension, implying that the number of supported edits can also scale exponentially. Since this bounds worst-case performance, practical performance can be even better, as demonstrated experimentally in Section 5. Empirically, we find that a single jet-pack may accommodate 10k edits with a negligible detector false positive rate (new Table 3), implying the performance ceiling is somewhat higher.
>
> **Detector Neuron**. Yes, in-place edits (and stealth attacks) use a very simple pruning strategy to select which neuron to replace. Empirically, this does not appear to cause problems and has the advantage of being cheap and needing no additional data. Another pruning strategy could be used without affecting the method. We will clarify Section B.1.3 and similar places in the revised version.
>
> **Difference with ROME**. The row/column mechanism provides a nonlinear switch for the edit, which enables our theoretical guarantees against damage to the model's performance. ROME's rank one update provides no such guarantees, since all edits respond linearly to all inputs, as discussed in Section 2 (please also see the related discussion of LoRA in response to Reviewer 3, question 1). Other work has observed that model performance breaks down as ROME-type edits are inserted, sometimes from a single bad edit - exactly what we aim to avoid. The new Table 2 shows the average impact of a single edit using ROME and our method. A further consideration is that our edits are trivially individually reversible/replaceable, which is not easily possible with rank one updates. Another difference is that we can directly test whether an edit is falsely responding to an input and act accordingly.
>
> **Limitations**. The edit success rate is described in detail in Section C.2 (lines 608--612) and is based on (1) the first token of the generated output matches the first token of the expected output (2) the target output is contained within the generated text. From examples, we can see this is sufficient to represent successful generations of the target outputs. We will clarify this in the revised version.

---

> > ### Comment · Reviewer_wwKs · 2024-08-11
> >
> > I thank the authors for their detailed response, which helped me to understand the motivation of the paper better. The authors have promised to include some further evaluations to address the concerns raised in the reviews. I am happy to revise my rating to a 6. I am looking forward to seeing the final version of the paper. Congratulations on your fine work!
> >
> > Why not an increased rating? - Although this paper makes novel contributions I don't see many practical applications as the edits lack generalization, which the authors have clarified that they were not aiming for. In my personal opinion (not meant to be a critique of the paper and the authors views), I think the authors are over-complicating the notion of generalization to some extent. Based on their example in the global response - the answer to *'Who is the Prime Minister of the UK?'* is a fact retrieval task for the LMs, the edit to which should indeed generalize well across different paraphrases of the prompt. For *'The year is 2015. Who is the Prime Minister of the UK?'* where the relation (according to ROME) can be considered as (s = "UK", r = "Prime Minister in year 2015") or (s = "Year 2015", r = "Prime Minister of UK"). And for the other example, *'Let's write a story about a world where everyone is called Blarg. .. '*, is about following cues from the in context prompt, which I think is a different task altogether.

---

> > > ### Author Response · Authors · 2024-08-12
> > >
> > > Thank you for responding to our comments, and for increasing the score. We would like to take this opportunity to clarify some remaining points.
> > >
> > > Regarding the question of generalising edits, we undoubtedly see the potential use for edits which really can reliably generalise. Unfortunately, current editing methods do not seem to be able to promise this. To take an example, we tried editing the phrase 'Who is the Prime Minister of the UK?' in GPT-J-6B using ROME, to produce the output 'Oliver' (for example). Afterwards, the edited model responded with 'Oliver' to all three of the example prompts, even when this is not an appropriate response because of the wildly different contexts.
> > >
> > > Of course, changing models or parameters may produce different results, but this demonstrates the lack of control and robustness these techniques offer with regards to generalisation. The re-formulations of the edit proposed by the reviewer demonstrate the need for nuanced extra knowledge about the query in order to implement a generalising edit, not required for our edits. Moreover, for editing a general purpose language model (as we aim to do), it is difficult to filter prompts at run time which have a different context as being 'out of scope'. This is especially true in the common setting where models have user-modifiable instructions contextualising prompts. As our experiments in the additional PDF attached to the global response confirm, even a single jet-pack (many can be added to a single model) can support exponentially many edits (in the feature dimension) without degrading the performance of the model. This provides an opportunity for the designer to fine-tune the applicability of their edits in a controlled fashion, accounting for the fact that the truthfullness of a response depends on the context in which a question is asked.

---

### Official Review · Reviewer_jzRE · 2024-07-11

**Soundness:** 3
**Presentation:** 4
**Contribution:** 3
**Rating:** 6
**Confidence:** 3

**Summary:**

This paper studies the problem of, when given a particular prompt, whether it is possible to surgically and efficiently edit a model’s parameters to produce a certain response in a way that does not otherwise change the behavior of the model. The paper provides an efficient technique for this in the form of both an in-place edit of a small amount of a model’s parameters, or alternatively as an additional block, “jet-pack”, inserted to encode the edits using the same framework. The key intuition of the technique comes from the perspective that the FFN block is a type of key-value memory (Geva et al. 2020), and we can rewrite a key that is not important to the overall model (smallest L1 norm, as used in this paper) to one that explicitly triggers on the desired input, and we can rewrite the corresponding value to a value that maximizes the likelihood of the desired output. The authors provide theoretical foundations that this change does not impact the general behavior of the model, as well as empirical evaluation of the technique over two factuality datasets to demonstrate the success rate of their technique, that it does not disturb the general ability of the model (on metrics like perplexity), and that the true false positive rate (FPR) is consistent with the FPR predicted by the theory.

**Strengths:**

1. This is a very well written paper that clearly lays out the problem being studied, introduces the techniques, theoretical foundations, and empirically verifies the theory and edit success.

1. The contribution is significant and novel: to edit the output of a model so precisely by purely adjusting two rows/columns in a MLP transformation is an excellent finding. The framework developed here is important and will be of interest for many in the field. It may spark other research works that can apply or extend this technique to address other open problems in LLMs (e.g. perhaps continual learning, model merging) .

1. Within the specific setting that the work studies, the paper is rigorous in its analysis and evaluation: modern LLMs of different architectures (Llama, GPT-J, Mamba) are evaluated, where to add the edit within the model is thoroughly ablated and plotted, methods and experiments cover a variety of scenarios: in-place and jet-pack editing, editing under certain contexts: corrupted prompt, unexpected context.

**Weaknesses:**

1. The motivation of the paper focuses on stealth, but this dimension could be better defined. The paper specifically studies "stealth" under the definition that 1) the edit is directly to the parameters and is somehow small 2) given a fixed set of prompts, only the edited prompts should show altered behavior. However, stealth could also be taken to mean: whether it is possible to detect in the weights that the model maliciously altered -- which this paper does not study.

1. It is unclear to what degree the jet-pack approach is "stealth" as it alters the parameters / layer stack used in the model.

1. Given the motivation of fixing hallucinations, the limitation of the approach is somewhat hidden by the set-up and assumptions. This may be perfectly fine if these limitations were clearly stated in the framing of the paper, however it does not seem to be and the results could be misinterpreted. Specifically:

    a. The mechanism of the edit is designed such that given a particular prompt, the model outputs a particular output. It seems like such an approach may only be effective under very specific prompts, which is not useful in the case of addressing hallucinations, which this paper uses as motivating example. If this is not true, it would be excellent to add additional empirical evidence that these edits can adjust concepts/facts rather than just the output given a specific prompt.

    b. If the edit is indeed specific and results replacing an entire row/column in the MLP, this would be limiting for real use cases.

    c. The edits are only done over the pre-trained models, and only for very short factual prompts. Perplexity ratio is only computed for 50 tokens, and over the datasets considered in this work. It is possible that the model may have compartmentalized Wikipedia / factual knowledge of this type in certain parts of the MLP, allowing for edits to be done in other parts of the MLP corresponding to other abilities. To demonstrate true "stealth" edit/attack that would deceive a standard user, would require evaluation that other capabilities are not impacted as well. Here, evaluation of perplexity over a general multi-domain dataset such as The Pile, or reporting performance a wide array of few-shot tasks would be much more convincing empirical evidence.

   d. In Section C.2, it is mentioned that edit success is defined as 1) the first token matches the first token of the expected output, and 2) the target output is contained within the generated text. This is different than Exact Match metrics typically used for close-book QA datasets like Natural Questions, TriviaQA, as it could allow for some degree of degeneration. Providing samples of outputs from the model could alleviate this concern.


Minor typos:
- Line 338

**Questions:**

1. In some ways the techniques here are similar to other parameter-efficient fine-tuning approaches. e.g. with LoRA [1] a low rank adapter is trained, and during inference its weights are added to the base model's weights. This is somewhat similar to the MLP key-value row/column trained here and replaced in the base model's weights. One could imagine training a LoRA adapter to address hallucinations while otherwise maintaining general performance. To what degree is this a work a stealth edit and such a standard approach is not? Is this a reasonable naive baseline to try?

1. What is the scaling like for the number of edits a model is able to take before degenerating?


[1] https://arxiv.org/abs/2106.09685

**Limitations:**

See weaknesses response above. The authors do provide a limitation section that covers a reasonable degree of limitations otherwise.

---

> ### Author Rebuttal · Authors · 2024-08-06
>
> We would like to thank the Reviewer for their thorough reading of the paper and detailed comments. We have responded to these individually below.
>
> **Weaknesses**
>
> 1. Here, we focus on stealth in the sense that the architecture is unchanged and/or performance on a large unknown validation set is unchanged. This is relevant to settings such as third-party foundation models used in a larger computational pipeline. Weight hashing can detect model changes, but would not indicate a stealth attack hidden among benign changes. Consider an analogue of the recent XZ hack: it could be very difficult to detect stealth attacks made by a trusted party over a long time period. It is vital that mechanisms for preventing and detecting stealth attacks are developed, work initiated by our theoretical tools.
>
> 2. A jet-pack is stealthy in the functional sense that it is virtually impossible to detect its impact on other model performance. Preserving the model's existing expected performance on repeated tasks is an important aspect of a model's reliability for its users. Jet-packs enable non-disruptive surgical fixes for identified problems.
>
> 3. a) We refer to our 'Global Rebuttal' discussion on generalising edits. In the revised version of the paper we will rephrase the introduction and augment the discussion section to clarify that we aim to correct 'specific hallucinations' (individual inputs which cause problems) while guaranteeing not to damage other model behaviour.
>
> b) Each in-place edit requires replacing a row/column of the perceptron matrices, to avoid changing the model structure. This in itself enables important real-world use cases, such as stealth attacks: just a single well-designed stealth attack hidden in a model could have catastrophic security implications. Jet-pack editing is more practical for large-scale editing, as additional rows and columns can be added or removed as necessary to make, remove or replace edits atomically and independently.
>
> c) As discussed in the 'Global Rebuttal', we specifically aim not to impact other capabilities of the model. Theorems 2 and 3 prove guarantees on this directly. In practice, two factors change the edited model's outputs (hence perplexity ratio): removing a neuron (for in-place editing), and edit detector false positives. Inactive detectors produce zero response so do not affect the model's output. We ran extra experiments to show the effect on perplexity of both sources. The Pile was withdrawn before March 2024 on ethical grounds so we instead used the Pile-10k dataset.
>
> The perplexity ratio from removing a neuron is shown in the new Figure 2 (see attached PDF), for 500 prompts from each of MCF, ZsRE and Pile-10k. For each, we calculate the perplexity ratio between the modified and original model on text generated by the models up to 50 tokens. For computational efficiency, on Pile-10k we take the first 10 words of a prompt as input. These results show that little perplexity is produced by setting a neuron to zero.
>
> The new Table 1 shows the FPR remains negligible even for cross-domain prompts. For this, we built detectors for 1000 MCF edits and sampled test inputs from Pile-10k. Consequently, we see that the additional perplexity on Pile-10k is comparable to the values reported on MCF and ZsRE in Section 5.
>
> d) We have provided example edited outputs from each model as Table 4 in the attachment, for inputs sampled from the MCF dataset. We are happy to provide more, but are limited for space in this response.
>
> **Questions**
>
> 1. We thank the reviewer for the opportunity to clarify this difference. Stealth edits could be seen as rank 1 updates to weight matrices as the edit only changes a single neuron. Major differences with LoRA include:
>
> a) *Learning a task vs a query*. LoRA was introduced and is primarily used to fine-tune and adapt existing pre-trained models to *new tasks*. These new tasks expect the LLM to process a large number of different queries, producing answers which differ from those of the original model. Our edits aim to alter the response to just one new specific query.
>
> b) *Information/knowledge and data for fine-tuning*. Unless the new task is known to live in a subspace orthogonal to previously learned tasks, LoRA requires data describing the new task and previously learned tasks to protect the fine-tuned model's performance on other tasks. Our edits do not require this. Instead, we give theoretical guarantees that responses of the edited LLM to other queries are unlikely to change.'
>
> c) *Performance guarantees*. As the LoRA paper states: 'The mechanism behind fine-tuning or LoRA is far from clear. How are features learned during pre-training transformed to do well on downstream tasks? ... We mostly depend on heuristics to select the weight matrices to apply LoRA to. Are there more principled ways to do it?' LoRA is 'learned' through sequential gradient updates of `low-rank weights'. Our method is a rank-1 one-shot learning followed by a light-touch gradient adaption to ensure that the model responds as intended to the target prompt. Importantly, the formulation of our method builds on inferrable geometric and statistical properties of the LLM's feature spaces. This enables theoretical assurances and performance guarantees.
>
> 2. For jet-packs (most appropriate for inserting many edits), the only cause of model degeneration is false detector responses to model inputs. Theorem 2 shows this is governed by the intrinsic dimension of the jet-pack's feature space. Empirically, we find that a single jet-pack may accommodate 10k edits with a negligible detector false positive rate (new Table 3), implying the performance ceiling is significantly higher.

---

> > ### Comment · Reviewer_jzRE · 2024-08-12
> > **Response**
> >
> > "For each, we calculate the perplexity ratio between the modified and original model on text generated by the models up to 50 tokens. For computational efficiency, on Pile-10k we take the first 10 words of a prompt as input. These results show that little perplexity is produced by setting a neuron to zero."
> >
> > Does this mean that perplexity in this work is calculated over model generations? E.g. text is generated by the model up to 50 tokens, then the model is used itself to score the text? Rather than having the model score the original document in the eval dataset?

---

> > > ### Author Response · Authors · 2024-08-13
> > >
> > > To clarify, the perplexity ratio metric aims to measure the extent to which the edits/attacks change the original behaviour of the model. This is in accordance with our overall aim of making targetted edits to a model, without impacting the model's broader baseline performance on a wide variety of tasks. For this reason, we measure the perplexity of the edited model to the outputs of the original model. We take the original model's perplexity to its own outputs as a baseline, which we use to produce the ratio. A perplexity ratio of 1 can therefore be used to indicate that the behaviour of the model has not changed for a given prompt, in combination with the range of complementary metrics given in Section 5.
> > >
> > > In the setting of this work, we feel that reporting raw model perplexities would not measure the ability of our edits to avoid disrupting the model. In particular, we use only public datasets and pre-trained models which have had their perplexity profiles thoroughly analysed elsewhere. Ultimately, it is therefore not the baseline performance of the model which is our focus here, but the ability of our edits to preserve it.

---

> > > > ### Comment · Reviewer_jzRE · 2024-08-13
> > > > **Updated rating**
> > > >
> > > > Thank you for your additional results and clarifications. In general I believe this work contains novel and valuable techniques that should be published, and that the right reading of this work is as a foundational one: it would set a valuable start for other works to potentially build on and improve. It seems that the authors are open to revising the framing as mentioned to be clearer and more specific about the aim of 'specific hallucinations' or 'specific edits'. I think being clearer upfront about the limitations of the work was one of my main concerns, as given the current framing it is easy to read the abstract and misconstrue the paper as a silver bullet for manipulating model parameters to fix hallucinations -- which it is not. I might argue that "hallucination" in general is an overloaded word that may imply some degree of generalization, that the authors might want to avoid in the abstract. Given the author's proposed revisions to the framing, I will raise my score to a 6.
> > > >
> > > > Why not higher? I think the author's extra experiments on ThePile-10k, and example outputs does add a little bit more confidence that the work does not adversely affect the model's abilities over various domains. However, when the authors define "stealth" as "stealth in the sense that the architecture is unchanged and/or performance on a large unknown validation set is unchanged," I think it is fair to empirically require that for most ways that a user might use or evaluate a checkpoint, that the performance / quality of the model is unchanged. For most practitioners this would mean performance on a wide set of few-shot tasks, or chat performance like MT-Bench, or at the very least, basic perplexity evaluation over documents, like the full ThePile-10k. The "stealth" part should from the user's perspective, rather than the author's perspective for the purposes of this research paper.
> > > >
> > > > For example, while it is reasonable for the paper to compare perplexity over the baseline model's outputs to see that the edited model still matches the baseline model -- it is much more likely for a user who is trying to verify that a model is the same to rerun normal perplexity evaluation to verify that it is reasonable, or even *identical*, in the strictest case, on the same hardware. This sort of comparison is a common sanity check when doing checkpoint conversion between platforms, or comparing hardware. The short sequence lengths considered in this work, also prevents us from understanding if the quality of the model would be affected at all for longer inputs / outputs.

---

> > > > > ### Author Response · Authors · 2024-08-14
> > > > >
> > > > > Thank you for your feedback and for updating the rating.
> > > > >
> > > > > We stress that it is difficult for us to run these additional experiments requested by the reviewer less than 24 hours before the end of the discussion period. Despite this, we have performed some further experiments which are described below, using gpt-j-6b float16 and the standard LM Evaluation Harness. Overall, these results demonstrate that our edits do not adversely affect the model's abilities over various domains, as requested by the reviewer. We will be happy to include these results into the appendix of the revised version.
> > > > >
> > > > > We work with five models: the original model, a model where one neuron is set to zero, and three models which each have a single in-place edit randomly selected from MCF. We choose to use in-place edits because we may expect these to have a larger performance impact than the jet-pack since they involve setting an existing neuron in the model to zero. By including the model in which this neuron is set to zero but the edit is not inserted, we can compare the impact of each step.
> > > > >
> > > > > 1. We measure perplexity of the models on the original document of the full Pile-10k dataset: the original model, the zero neuron model, and all three edited models had the exact same word and byte perplexity to at least four decimal places. This is supported by the fact that the FPR rates, as shown in Table 1 (of the PDF in the global response) for Pile-10k are low.
> > > > >
> > > > > 2. We measure performance scores on the few-shot tasks of the TinyBenchmarks (including 25-shot tinyARC, 5-shot GSM8K, 10-shot tinyHellaswag, 5-shot tinyWinoGrande). TinyHellaswag and tinyWinoGrande returned same accuracy (to at least four decimal places) for all five models, tinyGSM8K returned same Exact-Match score for both flexible-extract and strict-match filters for all five models (to the same precision), tinyARC returned 48.90% ACC for the original model, and 48.44% ACC for all edited models.
> > > > >
> > > > > 3. We measure performance scores via 5-shot on MMLU-Pro: the benchmark returned $0.1256\pm 0.003$ for the original model and $0.126\pm 0.003$ for the zero-neuron model and the three edited models with in-place edits.
> > > > >
> > > > > Note that for tinyARC and MMLU-Pro the zero-neuron model and the in-placed edited models shows the same accuracies. Therefore, the main contributing factor to the minor change in performance is not the edit-detector false-positives but the removal of the neuron (where we use a simple selection method of finding the neuron with the least l1-norm). Overall, our methods showed minimal impact on the original behaviour of model.
> > > > >
> > > > > We hope this extra experimental evidence resolves the reviewer's outstanding questions.

---

### Official Review · Reviewer_w4Ct · 2024-07-13

**Soundness:** 4
**Presentation:** 3
**Contribution:** 4
**Rating:** 7
**Confidence:** 4

**Summary:**

This paper proposes a new algorithm and studies a family of methods it refers to as *stealth edits*, which modify a large language model to selectively correct a set of known hallucinations without otherwise affecting the responses. It also proposes *intrinsic dimension*, a pairwise separability-based metric to determine the ease of editing any given network block of an LLM. It provides theoretical guarantees for the selectivity of edits and demonstrates that the probability of edit activation decreases exponentially with increasing intrinsic dimensionality.

It proposes two versions of the algorithm - one for in-place editing a network block to correct a given prompt and target output, and another for inserting an additional *jet-pack module* in the network to correct a series of hallucinations. Both of these algorithms optimize the rows/columns of the projection layers of the block with gradient descent to produce the expected target output(s). The *jet-pack block* controls the selectivity of an edit using a modified *RMSNorm* normalization layer that is re-centered to the mean normalized features of a set of Wikipedia prompts to maximize their intrinsic dimensionality.

It also highlights that these methods expose a potential vulnerability in LLMs and can potentially be used by malicious actors to perform a *stealth attack* to produce targeted responses for specific trigger inputs of their choice. Specifically, it discusses two variants of attacks - a *corrupted prompt attack* in which the attacker inputs slightly different versions of the trigger prompt and an *unexpected context attack* in which the attacker mentions the target response for only a given prompt that is followed by "context" sentence. In both of these cases, the attacks are difficult to monitor or track through automated tests.

It validates the effectiveness of their proposed algorithms by reporting the performance on MCF and ZsRE datasets using the metrics: edit success rate, perplexity ratio of attacked model and the original model, and actual/theoretical worse-case of detector false positive rates. It demonstrates a lower edit success rate and higher false positive rates in the earlier layers of the model in the case of stealth editing. It also shows a near-perfect detector false positive rate in the case of *jet-pack edits* and observes similar trends of selectivity in both variants of the stealth attack. The proposed algorithm's formulation unifies seemingly different related methods GRACE and Transformer-Patcher. It also notes that normalization functions in LLMs can help enable selective edits while also being a point of potential vulnerability in these models.

**Strengths:**

- This paper introduces a novel formulation and theoretical framework to analyze methods addressing the important problem of selectively correcting hallucinations. The formulation generalizes to a wide variety of architectures and selective editing algorithms.
- The metric *intrinsic dimensionality* seems particularly useful in understanding both the efficacy of selective edits and identifying potential vulnerabilities in an LLM.
- The experiments reveal important flaws in prior work such as the sub-optimal choices of applying edits at the end of the model and find them to be much more effective near the middle layers.

**Weaknesses:**

- The gap between theoretical expectations and empirical results w.r.t false positive rates in the layers towards the end isn't sufficiently addressed. The looser near-zero worst-case bounds for false-positive in this case also aren't that helpful.
- Some of the important observable trends in the figures of the experiments section aren't justified. Specifically, the trends for the edit success rate metric in Figures 3, 4, and 5 seem to vary widely across all layers among different model families, while it is relatively more consistent for the other metrics.

**Questions:**

1. Could you provide an intuitive or theoretical justification for why you observe the false positive rate slightly trend up towards the final layers of the model and find the optimal edit insertion location to be about half-way?

**Limitations:**

The authors have generally addressed the limitations and potential societal impacts of their proposed algorithm. A few suggestions to improve:
- In section [7], according to the guidelines they should provide further details on which error bars are used and state the corresponding confidence intervals in their experiments.
- While the negative social impacts of stealth attacks are discussed in different sections of the paper, it should ideally be discussed again in the Discussion section, especially the practicality or limitations of such attacks in practice.

---

> ### Author Rebuttal · Authors · 2024-08-06
>
> We would like to thank the Reviewer for their thoughtful comments on the paper. We have responded these individually to below.
>
> **Weaknesses**
>
> 1. Thank you for pointing this out. It is of course to be expected that the worst-case theoretical guarantees of Theorems 2 and 3 underestimate practical performance. Even in such cases, they still provide an indication of which areas of which areas of the model are better suited for editing.
>
>     We have also identified an unfortunately uninformative aspect of the original presentation in some of the figures. Rather than comparing the Theorem 3 worse case FPR to the detector false-positive rate on the wiki-test set, it is more informative to compare against the detector false-positive rate on potential trigger prompts (which the theoretical bounds offer guarantees on). This is clarified in the new Figure 1 (in the attached PDF), which enhances Figure 5 for the revised version. We have added FPR measures on potential trigger prompts, which demonstrates that our bounds are indeed informative. Similar changes will be made to clarify other figures.
>
> 2. The edit success rate measures whether our gradient descent algorithm was able to find an output vector from the edited block which produced the desired text response from the model. This explains why this metric behaves differently from the other metrics, which all instead involve properties of inputs to the block, in the form of model feature vectors. The results make it clear that additional work is required to improve the reliability with which output vectors are generated, although this was not a core focus of our research. Algorithms such as GRACE and ROME use similar methods to produce the desired outputs, so it is to be expected that progress on this will be transferable between methods.
>
> **Questions**
>
> 1. This is indeed an intriguing observation. From a theoretical perspective, this is because the intrinsic dimension of the feature vectors (in the sense of Definition 1) reaches a maximum around the middle of the model, before decreasing slightly towards the later layers. A lower intrinsic dimension corresponds to feature vectors which are more 'clumped together'. In early layers of the model, this could be because the features are predominantly encoding the last input token, while the features in later layers primarily encode the next output token. At the 'sweet spot' half way through the model, we hypothesise that the feature vectors encode more nuanced information about the complete text, which naturally requires a higher dimensional representation.
>
> **Limitations**
>
> 1. The shaded regions on relevant plots in Section 5 show the standard deviation in our results across many edits/attacks. To simplify the presentation, we report the maximum standard deviation for each model across all datasets. We intend to clarify this in the revised version. It is not clear to us how to meaningfully report confidence intervals when the distribution of the underlying data is unknown. That is why we resorted to showing standard deviation instead. We could provide max/minima, although we feel this will make the plots much more cluttered without adding much information.
>
> 2. We agree that this is an important societal risk which needs to be widely and fully discussed. When revising the article, we will add the following to the discussion section:
>
>     'The stealth attacks we have exposed here represent a new and potent threat to language models. They can be implemented without access to model training data and without fine-tuning, and are cheap enough that in many cases they can be implemented by a piece of malware. Their highly specific nature means they are very difficult to detect through conventional testing. Further investigation is therefore required to develop models which are resistant to stealth attacks (which may be guided by our intrinsic dimensionality metric), and alternative mechanisms to detecting their presence.'

---

> > ### Comment · Reviewer_w4Ct · 2024-08-12
> > **Rebuttal response**
> >
> > Thanks for answering the concerns and questions. I will keep the score.

---

### Official Review · Reviewer_ypKH · 2024-07-13

**Soundness:** 3
**Presentation:** 3
**Contribution:** 3
**Rating:** 7
**Confidence:** 4

**Summary:**

This paper focuses on stealth editing in large language models, presenting methodologies for making targeted, subtle changes to these models without retraining. The techniques, called "stealth edits," aim to correct specific issues like factual inaccuracies by directly updating the model's weights. The research also reveals that all modern language models are susceptible to stealth attacks, which are targeted and hard to detect, involving minimal changes to a model's weights to alter its response to a specific prompt. Experimental results demonstrate high success rates for both edits and attacks while maintaining low false positive rates. The findings have significant implications for AI system security and reliability, indicating that even extensively trained models can be vulnerable to targeted manipulation. The authors conclude by highlighting the broader impact of their work and the need for further research in this area.

**Strengths:**

1. This work analyzes how the intrinsic dimension of data is a crucial factor: the higher the dimension the higher the probability that a given edit is successful. The findings here are novel and intriguing, which could guide future research in the field. It also combines the edit and attack together.

2. The experiments are sufficient. They considered different architecture and different model sizes.

**Weaknesses:**

1. The paper's organization can be improved to make it easier to understand. Maybe the space is limited and I think some information can be moved from the Appendix to the main paper.

**Questions:**

1. In the text around Line 187-190, it says it would add a new row to W1 and a column to W2, and in Algorithm 2 Step 9， it still uses the W1 as the updated model. I'm a bit confused about whether the W1 here has extra rows. I didn't get this jet-pack editing here.

---

> ### Author Rebuttal · Authors · 2024-08-06
>
> We would like to thank the Reviewer for their comments and careful reading of the paper. We have responded to each comment below.
>
> **Weaknesses**
>
> Many thanks for the suggestion. In the submitted version, the Appendix contains details of the algorithmic steps, the experimental protocols, and the rigorous proofs of the theorems.
>
> In response to this comment, and minding the space limitation, we suggest that further material could be included at the end of Section 1 to briefly summarise key aspects of the Appendix; i.e. outlining the nature of the key algorithmic steps, experimental measures, and proof techniques. Further emphasis on the differences in meaning between the phrases 'stealth attack' and 'stealth edit' could be given by adding precise definitions to the text.
>
> **Questions**
>
> We thank the reviewer for pointing out this inconsistency. Algorithm 2 describes building a new jet-pack block encoding a fixed set of edits and adding it to a model, which involves building a new network block from scratch and so creates new W1 and W2 matrices. The text in the paragraph describes adding additional edits to an existing jet-pack (so adds rows/columns to the W1 and W2 matrices of the existing jet-pack). We propose to clarify this by changing the title of Algorithm 2 to 'Adding a new jet-pack block to correct multiple hallucinations', and replacing the first two sentences of the paragraph (roughly lines 187--189) with:
>
> 'A new jet-pack block to correct a given set of hallucinations is added to a model by constructing W1, W2, b and mu as described in Algorithm 2. Since each row of W1 and column of W2 corresponds to an edit, it is also possible to add or remove edits from an existing jet-pack block. Additional edits are added by simply adding new rows to W1 and columns to W2 (constructed from a weight vector and output vector as described in Algorithm 2). An edit may be removed by deleting the corresponding row and column.'

---

> > ### Comment · Reviewer_ypKH · 2024-08-10
> >
> > Thanks for your clarification.
> > I have no remaining concerns and will keep the score.

---

### Author Rebuttal · Authors · 2024-08-06

We would like to thank all of the reviewers for their detailed comments on the paper, which we have responded to individually. Two of the reviewers also raised an interesting philosophical discussion, regarding model edits which aim to generalise beyond the original target prompt. We discuss this topic in detail here, and plan to enhance the paper by briefly commenting on this in the discussion section.

As the reviewers identify, here we focus on the case of surgical edits to models to fix given individual prompts with the specific aim of guaranteeing not to damage (or otherwise alter) other functionality in the model. A range of other editing methods presented in the literature (and reviewed in Section 2) aim to edit models in ways that 'generalise' to also affect the model's outputs for other prompts. In various circumstances, however, such behaviour may not be desirable (a stealth attack is an immediate example of this), and we view the extent to which an individual edit should 'generalise' as fundamentally the user's choice. At present, there is a gap in the literature for editing methods which can guarantee to fix identified bugs without side effects, and for understanding the fundamental properties determining their success.

Model editing is ultimately a blunt tool compared with more expensive approaches such as fine-tuning. It is to be expected that as an edit generalises, it increases the risk of causing unintended model degeneration. This is further complicated by the inherent subtle complexity of natural language: should all similarly-phrased prompts also be considered hallucinations? What is the precise definition of a 'similarly-phrased prompt'? What about similar sentences in very different contexts? To take a standard model-editing example, consider: 'Who is the Prime Minister of the UK?' and 'The year is 2015. Who is the Prime Minister of the UK?' or 'Let's write a story about a world where everyone is called Blarg. Who is the Prime Minister of the UK?', etc. Moreover, it is often observed that (semantically) small changes to prompts can significantly change model outputs (the phenomenon exploited by adversarial attacks, for example). It is not difficult to imagine scenarios where overly-generalising edits introduce unknown new errors of their own. Preventing edited models from catastrophically forgetting previous learning is in itself an important alternate form of generalisation, as employed in the field of life-long learning. This is particularly important in tasks where prompts may be highly structured: the risks of over-generalisation would be borne without any counterbalancing advantage.

Many empirical studies, such as reference [15] introducing the GRACE method (which may be efficiently implemented using a jet-pack as shown in Section 7), suggest these editing methods may be able to generalise to rephrased prompts. As with all empirical work, however, the quality of the conclusions drawn from these experiments relies on the quality of the benchmark dataset. Results also depend on the choices made when quantifying generalisation. The datasets typically used for such tests present just a few phrasings similar to the original prompt; is this sufficient to certify that an edit will generalise as desired, or simply that some additional model behaviour is affected by the edit? This is a common theme: studies investigating editing mechanisms which generalise to rephrased prompts appear to be universally empirical in nature, and even a compelling rigorous definition of 'generalisation' in the context of natural languages is absent from the literature. Understandably, the notion of generalisation in the context of language modelling is very different from generalisation within the framework of classical statistical learning theory. We see a need, therefore, for an alternative theoretical framework through which we are able to systematically study edit generalisation -- a nontrivial task which we plan to address in future work.

From an operational perspective, the task of building generalising edits is also nontrivial. Collecting high-quality re-phrased prompts and corrected responses through which to construct and verify/test each edit (or accurately identifying subject-object-relationship triples in the case of frameworks such as ROME [21]) requires time-consuming manual intervention. The targeted approaches we consider here could be used to insert edits automatically in response to user feedback. Follow-up work could consider approaches to later 'merge' or combine collections of related edits, perhaps during the next round of fine tuning. We also note that from an attack perspective, specificity is highly desirable as it makes detection more difficult.

We hope that this global response explains the motivation and the intended impact of our work, and clarifies the difference between the approach and settings we proposed in this work and what has been done so far in the literature on editing LLMs.

---

### Decision · Program_Chairs · 2024-09-25

**Decision:**

Accept (poster)

**Comment:**

The paper studies the problem of selectively editing a LM for a very specific fact without retraining.  It presents an efficient method exploiting the model high intrinsic dimensionality. The method is useful to correct hallucinations but it also exposes a new type of modelvulnerability.

The reviewers agreed that the paper is novel and that it is presenting an interesting problem. They had some doubts, related to the motivation of the paper and a more precise problem definition, as well as concerns with the distance between theoretical and empirical results. However, after the discussion, the reviewers had a more positive opinion of the paper, and they agreed that, while its scope might be more limited than what originally implied by the authors, it can be seen as a "foundational" paper defining an original problem.